

# On currents in the $O(n)$ loop model

Jesper Lykke Jacobsen[1,2,3], Rongvoram Nivesvivat[4] and Hubert Saleur[1,5]

**1** Institut de physique théorique, CEA, CNRS, Université Paris-Saclay
**2** Laboratoire de Physique de l'École Normale Supérieure, ENS,
Université PSL, CNRS, Sorbonne Université, Université de Paris
**3** Sorbonne Université, École Normale Supérieure, CNRS, Laboratoire de Physique (LPENS)
**4** Yau Mathematical Sciences Center, Tsinghua University, Beijing, 100084 China
**5** Department of Physics and Astronomy, University of Southern California, Los Angeles

## Abstract

Using methods from the conformal bootstrap, we study the properties of Noether currents in the critical $O(n)$ loop model. We confirm that they do not give rise to a Kac-Moody algebra (for $n \neq 2$), a result expected from the underlying lack of unitarity. By studying four-point functions in detail, we fully determine the current-current OPEs, and thus obtain several structure constants with physical meaning. We find in particular that the terms $:J\bar{J}:$ in the identity and adjoint channels vanish exactly, invalidating the argument made in [1] that adding orientation-dependent interactions to the model should lead to continuously varying exponents in self-avoiding walks. We also determine the residue of the identity channel in the $JJ$ two-point function, finding that it coincides both with the result of a transfer-matrix computation for an orientation-dependent correlation function in the lattice model, and with an earlier Coulomb gas computation of Cardy [2]. This is, to our knowledge, one of the first instances where the Coulomb gas formalism and the bootstrap can be successfully compared.



# 1  Introduction

It is a well-known fact that ordinary two-dimensional critical statistical-mechanics systems with continuous symmetries are described in the continuum limit by Wess-Zumino-Witten (WZW) models on the corresponding group. While we are not aware of a rigorous proof of this result, it has been widely used, starting with the analysis of the XXX spin chain and the corresponding $O(3)$ sigma model at $\theta = \pi$ [3], and has been a cornerstone of the discussions around the continuum limit of the integer quantum Hall plateau transition (see [4] and references therein). The existence of conserved charges leads, via the Noether theorem, to the existence of local currents whose conformal dimensions are not renormalized [5], and which are of the form $(\Delta, \bar{\Delta}) = (1, 0)$ and $(0, 1)$. With unitarity, this is enough to assert that these currents are really chiral, and the existence of an underlying Kac-Moody algebra follows.

Without unitarity, however, it is not guaranteed that the local currents are purely chiral, as it could well be that their derivatives are states with zero norm-square which do not actually vanish (i.e., they have non-zero scalar products with some other fields of the theory and hence cannot be eliminated from the problem). In fact, it was argued long ago [6] that the conformal field theory (CFT) for $O(n)$ loop models is *not* a WZW theory. This followed from a simple counting argument: in the partition functions, the degeneracy of fields with weights $(1, 0)$ and $(0, 1)$ is indeed the dimension of the adjoint $d_{[11]}$, but the degeneracy of fields with weights $(1, 1)$ is significantly smaller that $d_{[11]}^2$, indicating that chiral and anti-chiral components are not independent—a fact that is possible only in the presence of logarithmic terms. A well-

known example of such a situation is provided by symplectic fermions [7]—which turn out to be relevant to the case of the $O(n)$ model with $n = -2$; see below. The first purpose of this paper is to find out what happens in the $O(n)$ loop model for arbitrary values of $n$ in the critical domain, using in particular the bootstrap techniques recently developed in [8].

The fate of the currents in this model is more than an anecdotical question: it also requires great caution when importing arguments familiar from unitary situations. It was suggested [1, 2, 9] for instance, using such arguments, that in self-avoiding walks (related with the limit $n \to 0$ of the $O(n)$ model), orientation-dependent interactions, which can be formulated within the CFT as $J\bar{J}$ perturbations, would lead to continuously varying exponents. This prediction was never borne out by numerical studies (see, e.g., [10] for a thorough review), and remains a mysterious discrepancy between theory and simulations in a field otherwise thoroughly understood. We will see below how it can be explained using the current-current Operator Product Expansions (OPEs) derived from the bootstrap.

The paper is organized as follows. We start in Section 2 by reviewing the spectrum of the critical $O(n)$ loop model. In Section 3 we discuss general features of the current-current OPE derived from the bootstrap, and we show that they fail to form a Kac-Moody algebra. In Section 4 we analyze the details of the current-current OPE, and show how the currents, while not obeying Kac-Moody algebra relations, remain compatible with the existence of a global, non-chiral $O(n)$ symmetry. We also determine in this section the residue of the leading singular term in the current-current OPE (the "level" parameter $k$). Remarkably, the result— obtained using the bootstrap and its recent analytical solution—agrees with the one obtained by Cardy [9] using Coulomb Gas (CG) techniques. In this section we also investigate numerically various aspects of the current-current OPE, using bootstrap computations, and find excellent agreement with the theoretical predictions. In Section 5 we examine the implications for two limits, $n \to \pm 2$, in which the $O(n)$ model can be related to free-field theories. Finally, in section 6 we discuss the application of our results to loop models, and we relate the parameter $k$ to a correlation function involving oriented loops. In particular, in section 6.1 we revisit the argument of [1] and point out explicitly how it fails once the proper structure of the OPE is taken into account. Applications and generalizations are discussed in the conclusion. Appendix A provides more technical information about the bootstrap solutions. In Appendix B we obtain $k$ numerically from the lattice model, by a transfer matrix calculation that exploits the link established in Section 6, finding again agreement with the analytical predictions.

## 2 Spectrum of the model

We provide a review on the spectrum of the critical $O(n)$ loop model, as well as analyticity in the model's parameters, such as the central charge and the loop weight $n$. This section also serves as an introduction to our notations.

### 2.0.1 The $O(n)$ model and its parameters

The $O(n)$ loop model is an ensemble of non-intersecting loops on the hexagonal lattice, wherein we assign the weight $n$ to each loop and the fugacity $K$ to each monomer, see appendix B for detail on the lattice model. The model is known to become critical at the values of $K$ given in (B.3). At the critical value $K = K_c = (2 + \sqrt{2-n})^{-\frac{1}{2}}$, the $O(n)$ loop model exhibits a continuous phase transition where the two-point functions change from an exponential decay with the distance to a power-law decay. This fixed point is known as the *dilute fixed point*. At the other fixed point in (2), the so-called *dense fixed point*, the two-point functions remain algebraic, but their critical exponents change.

What we call the $O(n)$ CFT is a conformal field theory that describes the continuum limit of the critical $O(n)$ loop model. It was defined and studied in [8]. The central charge $c$ of the $O(n)$ CFT is related to the loop weight $n$ as follows [11]:

$$c = 13 - 6\beta^2 - 6\beta^{-2}, \quad \text{and} \quad n = -2\cos(\pi\beta^2), \quad \text{for} \quad \Re(\beta^2) > 0. \tag{1}$$

The constraint on $\beta^2$ arises from the condition that correlation functions converge [12]. Values of particular physical interest are $-2 \le n \le 2$, for which the dilute and dense phases are obtained by choosing

$$\text{dilute}: \beta^2 \in [1, 2] \Longleftrightarrow c \in [-2, 1], \tag{2a}$$

$$\text{dense}: \beta^2 \in (0, 1] \Longleftrightarrow c \in (-\infty, 1]. \tag{2b}$$

In other words, the CFTs describing each phase in (2) are special cases of the $O(n)$ CFT. There also exists another critical point at $K = \infty$ where the ensemble of loops becomes fully packed [13], in the sense that each lattice vertex is traversed by one loop. This distinct critical point is beyond the scope of this paper and belongs to another universality class that is expected to be described by a CFT with higher symmetry [14].

### 2.0.2 $O(n)$ representations and conformal dimensions

The CFT contains a set of primary fields (local operators) that transform in representations of the model's symmetry. Since the $O(n)$ CFT possesses formally a global $O(n)$ symmetry for generic $n$ [15], the spectrum of the model is a set of primary fields which transform both in representations of $O(n)$ and of conformal symmetry. For generic $n$, $O(n)$ representations can be parametrized by Young diagrams of arbitrary size, and we shall write these Young diagrams as decreasing sequences of positive integers. For example,

$$[\,]: \quad \bullet, \quad [2]: \quad \square\square, \quad [11]: \quad \begin{smallmatrix}\square\\\square\end{smallmatrix}, \quad [5421]: \quad \text{.} \tag{3}$$

On the other hand, Virasoro representations are labelled by conformal dimensions, which can be conveniently parametrized by the Kac indices,

$$\Delta_{(r,s)} = \frac{c-1}{24} + P_{(r,s)}^2, \quad \text{with} \quad P_{(r,s)} = \frac{1}{2}\left(r\beta - s\beta^{-1}\right). \tag{4}$$

### 2.0.3 Spectrum

The spectrum of the $O(n)$ CFT was first obtained via the torus partition function computed by the Coulomb gas technique in [11]. With our conventions, we have[1] [11,17],

$$\mathcal{S}^{O(n)} = \{V_{(1,s)}^D\}_{s \in 2\mathbb{N}^*+1} \cup \{V_{(r,s)}^\lambda\}_{r \in \frac{1}{2}\mathbb{N}^*, s \in \frac{\mathbb{Z}}{r}}. \tag{5}$$

The degenerate fields $V_{(1,s)}^D$ have left and right conformal dimensions $(\Delta_{(1,s)}, \Delta_{(1,s)})$ and are thus diagonal. They transform as the singlet under $O(n)$ symmetry and have multiplicity 1. The fields $V_{(r,s)}^\lambda$ have conformal dimensions $(\Delta_{(r,s)}, \Delta_{(r,-s)})$ and are thus in general non-diagonal (see below). They transform in the $O(n)$ representations $\lambda$, and have in general non-trivial multiplicities (for example, the field $V_{(\frac{5}{2},0)}^{[21]}$ has multiplicity 2). The analytic expressions of the

---

[1]In the dense case, our choice of parametrization for conformal weights leads, since $\beta^2 \in (0,1]$, to conformal dimensions $\Delta_{(r,s)}$ where the $r$ and $s$ labels are interchanged as compared to the conventions of the classical references [11,16].

multiplicity of $V_{(r,s)}^{\lambda}$ can be written as a complicated combination of polynomials in $n$ and can be found in [8]; we refrain from repeating them here.

As previously mentioned, the degenerate fields $V_{\langle 1,s\rangle}^{D}$ transform in degenerate representations of the Virasoro algebra. The non-diagonal fields $V_{(r,s)}^{\lambda}$ may transform either in Verma modules or the logarithmic representations described in [18], depending on their Kac indices. The summary of how Virasoro and $O(n)$ symmetries act upon the spectrum (5) can be found in [8,17]. We refrain from repeating these results here, but it is still useful to display, nonetheless, a few examples of how non-diagonal primary fields with the dimensions $(\Delta_{(r,s)}, \Delta_{(r,-s)})$ transform under $O(n)$ symmetry. For selected cases $(r,s)$ with $r \leq 2$, we find the following $O(n)$ decompositions [17]:

$$(1/2, 0) : [1], \tag{6a}$$
$$(1, 0) : [2], \tag{6b}$$
$$(1, 1) : [11], \tag{6c}$$
$$(2, 0) : [4] + [22] + [211] + [2] + [\,], \tag{6d}$$
$$(2, 1/2) : [31] + [211] + [11], \tag{6e}$$
$$(2, 1) : [31] + [22] + [1111] + [2]. \tag{6f}$$

The action of $O(n)$ symmetry on $(\Delta_{(r,s)}, \Delta_{(r,-s)})$ and $(\Delta_{(r,s')}, \Delta_{(r,-s')})$ coincides for $s - s' \in 2\mathbb{Z}$: therefore, it is sufficient to show the results for $0 \leq s \leq 1$. Representation labels will be kept implicit unless otherwise needed. Currents, for instance, will often be denoted as $J, \bar{J}$, keeping implicit that they transform in the adjoint $[11]$ and therefore come with multiplicity $d_{[11]}$. When a label is needed, as e.g. when writing the OPEs explicitly, we will use uppercase Latin letters $A, B, \ldots$ for the adjoint $(J^{A}, \bar{J}^{A})$ and lower-case Latin letters $a, b, \ldots$ for the fundamental.

### 2.0.4 Diagonal versus non-diagonal

In our conventions, the case $s = 0$ in the second component of (5) has zero conformal spin, but we still refer to this case as non-diagonal: more precisely, our definition for a non-diagonal field is a field whose fusion product with the degenerate field $V_{\langle 1,s\rangle}^{D}$ gives only non-diagonal fields. For readers more familiar with the early works on this problem, the $2r$-leg "fuseau" or "watermelon" field has conformal dimensions $\Delta = \bar{\Delta} = \Delta_{(r,0)}$. The energy field—coupled to the fugacity of edges in the lattice model—is the diagonal field with $\Delta = \bar{\Delta} = \Delta_{\langle 1,3\rangle}$. Further reminders about the underlying lattice model and the relationship with the $O(n)$ CFT will be given below (see also the introduction in the paper [8]).

## 3 The current-current OPEs

The definition of the model requires considering $n$ as a continuous variable and raises questions about the precise meaning of "$O(n)$ symmetry", in particular its consequences on the properties of the CFT. Recent work on giving a categorical interpretation to the model [15] leads to the conclusion that Noether's theorem should still apply, and therefore that there should be in the CFT a pair of local fields with conformal weights $(1, 0)$ and $(0, 1)$, respectively, obeying a local form of charge conservation. Indeed, it is well known [6,19] that the $O(n)$ CFT admits a pair of primary fields

$$J = V_{(1,-1)}^{[11]}, \quad \bar{J} = V_{(1,1)}^{[11]}, \tag{7}$$

with the correct conformal weights. Both $J$ and $\bar{J}$ (which will be referred to from now on as currents) transform in the adjoint representation [11]. We will, whenever necessary, indicate this by denoting the current components as $J^A, \bar{J}^A$, where the label $A$ takes values in [11].

In contrast to the case of WZW models, however, the currents $J$ and $\bar{J}$ of the $O(n)$ CFT are not holomorphic (anti-holomorphic). Rather, they belong to indecomposable representations of the Virasoro algebra [18,19]. Charge conservation—which will be discussed in more detail in Section 3.3—imposes the constraint

$$\bar{\partial} J = \partial \bar{J}\,, \tag{8}$$

but, crucially, none of these two terms vanishes. Consequently, the current-current OPEs in the $O(n)$ CFT will have to mix in general $z$ and $\bar{z}$ coordinates, as we shall see explicitly below.

For equation (8) to hold without each term being separately zero, $\bar{\partial} J$ and $\partial \bar{J}$ must be the same diagonal primary field of dimensions $(1,1)$ and zero norm-square: a non-vanishing level-one null vector that belongs to the two modules generated by $J$ and $\bar{J}$ [20]. How the current-current OPEs differ from those of a WZW model and what the corresponding physical consequences are is one of the main concerns of this paper. For now, let us start with some general features of the current-current OPEs. We start by recalling the tensor product, for generic $n$, of two adjoint $O(n)$ representations:

$$[11] \times [11] = [1111] + [211] + [22] + [2] + [11] + [\,]\,. \tag{9}$$

The next step is to write down the spectra of the fusion products $JJ$ and $J\bar{J}$, where the spectra of $\bar{J}\bar{J}$ and $\bar{J}J$ are the same as those first two. These fusion products were obtained in [8] by numerically bootstrapping the current four-point function, while also taking into account the $O(n)$ tensor product (9). The result is

$$JJ \sim \sum_{\lambda \in [11] \times [11]} \sum_{k \in \mathcal{S}^{[\,]}} V_k^\lambda\,, \tag{10a}$$

$$J\bar{J} \sim \sum_{k \in \mathcal{S}^{\lambda} - \langle 1,s \rangle^D} V_k^{[\,]} + \sum_{\lambda \in [11] \times [11] - [\,]} \sum_{k \in \mathcal{S}^\lambda} V_k^\lambda\,, \tag{10b}$$

where $\mathcal{S}^\lambda$ denotes the set of Kac indices for primary fields transforming in the $O(n)$ representations $\lambda$. The sets $\mathcal{S}^\lambda$ can be found in the equations (4.2)–(4.7) and (4.27) of [8]. Furthermore, observe that $O(n)$ symmetry allows the $O(n)$ singlets to propagate in $J\bar{J}$ since both fields transform as the $O(n)$ adjoint representation. However, conformal symmetry forbids the conformal singlets—the identity field $V_{\langle 1,1 \rangle}^D$ and the other degenerate fields in (5)—from appearing in $J\bar{J}$. Let us also point out here that the multiplicities of the fields appearing on the right-hand sides of (10a) and (10b) are not known in general: this issue still remains an open problem.

To see some other general features of the current-current OPE, we will compute explicitly the leading terms of $JJ$, whereas results for $J\bar{J}$ and $\bar{J}\bar{J}$ will be written down immediately from those for $JJ$, using the degenerate shift equations of $V_{\langle 1,s \rangle}^D$, as will be shown in (18) and (13).

## 3.1 Case of $JJ$ and $\bar{J}\bar{J}$

First, we introduce the two-point and three-point structure constants $B_V$ and $C_{V_1 V_2 V_3}$:

$$\langle V_1(z_1, \bar{z}_1) V_2(z_2, \bar{z}_2) \rangle = \delta_{12} B_{V_1} z_{12}^{-\Delta_1} \bar{z}_{12}^{-\Delta_1}\,, \tag{11a}$$

$$\left\langle \prod_{i=1}^{3} V_i(z_i, \bar{z}_i) \right\rangle = C_{V_1 V_2 V_3} \mathcal{F}^{(3)}(\Delta_i, z_i) \mathcal{F}^{(3)}(\bar{\Delta}_i, \bar{z}_i)\,, \tag{11b}$$

where we denote $z_{ij} := z_i - z_j$. For compactness, we have introduced the function

$$\mathcal{F}^{(3)}(\Delta_i, z_i) = z_{12}^{\Delta_3 - \Delta_1 - \Delta_2} z_{13}^{\Delta_2 - \Delta_1 - \Delta_2} z_{23}^{\Delta_1 - \Delta_2 - \Delta_3} . \tag{12}$$

In principle, the OPEs $JJ$ and $\bar{J}\bar{J}$ share the same qualitative features, since the OPE coefficients of $JJ$ and $J\bar{J}$ are related by the shift equations

$$\frac{C_{JJV}}{C_{\bar{J}\bar{J}V}} = \begin{cases} \prod_{\sigma, \eta = \pm} (P_{(1,\sigma)}^2 - P_{(r,\eta s)}^2)^\eta, & \text{for} \quad V = V_{(r,s)}^\lambda, \\ \\ 1, & \text{for} \quad V = V_{\langle 1,1 \rangle}^D . \end{cases} \tag{13}$$

The relations (13) can be obtained by studying OPE coefficients of $J$ and $\bar{J}$ in the $s$-channel of $\langle JVJV \rangle$ and $\langle \bar{J}V\bar{J}V \rangle$. Therefore it is sufficient to discuss only the results for $JJ$. We will focus on the leading terms of this OPE, which can be obtained in a pedestrian way by solving Ward identities systematically. Similar computations can be found in the book [21] and we will only display the results here. Let us start with the leading terms in the singlet channel of $JJ$. We normalize the two-point function of the identity field $V_{\langle 1,1 \rangle}^D$ to be 1, and find

$$J(z, \bar{z})J(0)\Big|_{[\,]} \sim k\left\{V_{\langle 1,1 \rangle}^D + \frac{2}{c}z^2 T(z) + \frac{2}{c}\bar{z}^2 \bar{T}(\bar{z})\right\} + \frac{C_{JJV_{(2,0)}^{[\,]}}}{B_{V_{(2,0)}^{[\,]}}}|z|^{2\Delta_{(2,0)}}V_{(2,0)}^{[\,]}$$

$$+ |z|^{2\Delta_{(2,2)}}\left\{\bar{z}^2 \frac{C_{JJV_{(2,2)}^{[\,]}}}{B_{V_{(2,2)}^{[\,]}}}V_{(2,2)}^{[\,]} + \frac{C_{JJ(L_0 - \Delta_{(2,-2)})W_{(2,2)}^{[\,]}}}{B_{(L_0 - \Delta_{(2,-2)})W_{(2,2)}^{[\,]}}}|z|^2 \log|z|(L_0 - \Delta_{(2,-2)})W_{(2,2)}^{[\,]}\right\} + \dots, \tag{14}$$

where the notation $\sim$ means that overall scale factors—such as those written out in (11)—have been omitted for convenience, whereas $T(z), \bar{T}(\bar{z})$ denote the usual components of the stress-energy tensor, and we recall that $V_{(2,2)}$ and $V_{(2,0)}$ have the same $O(n)$ decomposition. Furthermore, we have also introduced the parameter $k$ as a shorthand for the two-point structure constants of the currents,

$$k = B_J = B_{\bar{J}} . \tag{15}$$

How to determine exact formulae for the leading three-point and two-point structure constants in the current-current OPEs will be discussed in Section 4.

The striking feature of (14) is that the OPE is not only non-holomorphic but also involves some logarithmic dependency on the coordinates $z$ and $\bar{z}$. The latter property follows from the fact that some of primary fields on the right-hand side of (10a) have non-vanishing null vectors, which in turn have logarithmic partners. For example, $V_{(2,2)}$ has a non-vanishing level-two null vector $(L_0 - \Delta_{(2,-2)})W_{(2,2)}^{[\,]}$ that comes with a logarithmic partner $W_{(2,2)}$ [18], whose coefficient can be completely determined by using only conformal symmetry, however we refrain ourselves from writing it down explicitly. Next, we consider the adjoint channel of $JJ$, wherein $J$ itself appears:

$$J(z, \bar{z})J(0)\Big|_{[11]} \sim \frac{C_{JJJ}}{k}\left\{zJ + \frac{1}{2}z^2 \partial J + \frac{\beta^2 - 1}{\beta^2 + 1}\left(\bar{z}\bar{J} + \frac{1}{2}\bar{z}^2 \bar{\partial}\bar{J}\right)\right\}$$

$$+ \frac{C_{JJV_{(2,\frac{1}{2})}^{[11]}}}{B_{V_{(2,\frac{1}{2})}^{[11]}}}z^{\Delta_{(2,\frac{1}{2})}}\bar{z}^{\Delta_{(2,-\frac{1}{2})}}V_{(2,\frac{1}{2})}^{[11]} + \frac{C_{JJV_{(2,-\frac{1}{2})}^{[11]}}}{B_{V_{(2,-\frac{1}{2})}^{[11]}}}z^{\Delta_{(2,-\frac{1}{2})}}z^{\Delta_{(2,\frac{1}{2})}}V_{(2,-\frac{1}{2})}^{[11]} + \dots \tag{16}$$

From (7), the second Kac indices of $J$ and $\bar{J}$ considered as primary fields differ by 2, and thus the ratio between three-point structure constants $C_{JJJ}$ and $C_{JJ\bar{J}}$ is fixed by the degenerate shift equation of the degenerate field $V_{\langle 1,s\rangle}^D$. We find

$$\frac{C_{JJ\bar{J}}}{C_{JJJ}} = \frac{\beta^2 - 1}{\beta^2 + 1}, \tag{17}$$

which yields the coefficient of $\bar{J}$ in (16). Furthermore, from direct computation, we also find that the logarithmic partner $W_{(1,1)}$ of $\bar{\partial}J$ and $\partial\bar{J}$ decouples from the OPE (16). From our previous work, this decoupling of $W_{(1,1)}$ from the OPE $JJ$ had also been observed at the level of correlation functions: the logarithmic conformal block of $J$ in the current four-point function $\langle JJJJ\rangle$ becomes accidentally non-logarithmic due to cancellations of singularities in the conformal block. See Section 3.1 of [8] for a detailed discussion.

From (16)–(17), it is then clear that the $JJ$ OPE is non-holomorphic for generic $\beta^2$. However, observe that, in the limit $\beta^2 \to 1$, $\bar{J}$ and its descendants decouple from $JJ$. Therefore we expect $JJ$ to be holomorphic in this special case (and this case only), as will be discussed in Section 5.1.

## 3.2 Case of $J\bar{J}$

As previously mentioned, the OPE $J\bar{J}$ can be obtained directly from $JJ$, since these two OPEs are related by the degenerate shift equation of $V_{\langle 1,s\rangle}^D$, which reads

$$\frac{C_{JJV}}{C_{J\bar{J}V}} = \begin{cases} \displaystyle\prod_{\eta=\pm}(P_{(1,-\eta)}^2 - P_{(r,\eta s)}^2)^\eta, & \text{for} \quad V = V_{(r,s)}^\lambda, \\[12pt] 0, & \text{for} \quad V = V_{\langle 1,1\rangle}^D. \end{cases} \tag{18}$$

Notice that the special case (17) can be recovered from this by considering the first of the above relations (18) as the limit $\epsilon \to 0$ of $V = V_{(1,-1+\epsilon)}$. One of way of deriving (18) is by considering the OPE coefficients of $J$ and $\bar{J}$ in the $s$-channel of $\langle JVJV\rangle$ while assuming the normalization (15). Notice also that the shift equations (18) are independent of the $O(n)$-label $\lambda$, since these relations are consequences of conformal symmetry only.

For generic $n$, the set of equations (18) implies that the spectra of the OPEs $J\bar{J}$ and $JJ$ only differ by the degenerate fields $V_{\langle 1,s\rangle}^D$ in $JJ$. In other words, the singlet channel of $J\bar{J}$ reads

$$J(z,\bar{z})\bar{J}(0)\Big|_{[\,]} \sim \frac{C_{J\bar{J}V_{(2,0)}}}{B_{V_{(2,0)}}}|z|^{2\Delta_{(2,0)}}V_{(2,0)} + \dots, \tag{19}$$

wherein $V_{\langle 1,s\rangle}^D$ are absent. For the adjoint channel, using (17), we find

$$J(z,\bar{z})\bar{J}(0)\Big|_{[11]} \sim \frac{\beta^2 - 1}{\beta^2 + 1}\left(zJ + \frac{1}{2}z^2\partial J\right) + \frac{(\beta^2 - 1)^2}{(\beta^2 + 1)^2}\left(\bar{z}\bar{J} + \frac{1}{2}\bar{z}^2\bar{\partial}\bar{J}\right) + \dots \tag{20}$$

Observe that neither $J$ nor $\bar{J}$ appears in $J\bar{J}$ at $n = 2$, where we expect the $O(n)$ currents to become holomorphic. We will elaborate on this special case in Section 5.1.

## 3.3 No Kac-Moody algebra

As discussed in the introduction, one of our goals is to study more precisely how the currents in our model—which enjoy a local, continuous symmetry (albeit at the price of a continuation in $n$, or a more categorical description)—end up *not* giving rise to a Kac-Moody algebra.

Obviously, the OPEs $JJ$ and $J\bar{J}$ in the $O(n)$ CFT are quite different from those in WZW models, in particular because of the non-chiral terms which we saw in (14) and (16). At first sight, these OPEs are rather close to the current-current OPEs of the deformed supergroup WZW models [22, 23], which are also non-chiral. However, the OPEs in these references only involve integer powers of $z$ and $\bar{z}$, whereas in our case we also have non-integer powers. (Maybe this is because the OPEs in [22, 23] are only valid to leading order in perturbation theory.) Also, we find logarithmic terms in both the $JJ$ and $J\bar{J}$ OPEs, whereas logarithmic terms in [22, 23] only occur in $J\bar{J}$.

To make things more concrete, we shall focus in what follows on only a few terms, with the specific goal of connecting our computations with physical observations and the global symmetry of the model. We have the general structure

$$J^A(z,\bar{z})J^B(w,\bar{w}) = \frac{k\delta^{AB}}{(z-w)^2} + f_C^{AB}\left[\lambda\frac{J^C}{z-w} + \mu\frac{\bar{z}-\bar{w}}{(z-w)^2}\bar{J}^C\right] + \dots, \tag{21a}$$

$$J^A(z,\bar{z})\bar{J}^B(w,\bar{w}) = f_C^{AB}\,\nu\left[\frac{J^C}{\bar{z}-\bar{w}} + \frac{\bar{J}^C}{z-w}\right] + \dots \tag{21b}$$

Here the labels $A, B, C$ run over the adjoint representation [11] of dimension $n(n-1)/2$, and $f_C^{AB}$ are the $O(n)$ structure constants, which take values 0 or $\pm 1$. This will be discussed in detail in Section 6. In (21), the dots stand for non-chiral terms, some of which also depend logarithmically on $|z|$, as we have seen in (14). Furthermore, the coefficients $k, \lambda, \mu, \nu$ are all well defined up to a global multiplicative factor to be discussed below.

### 3.3.1 Zero-mode algebra

We shall mostly be interested in the algebra of zero modes that results from (21), and thus turn to equal-time commutators, following [22, 23]. To proceed, it is more convenient to switch from the complex coordinates, $z$ and $\bar{z}$, to time and spatial coordinates, $\tau$ and $\sigma$. Setting $z = \sigma + i\tau$ and $\bar{z} = \sigma - i\tau$, the equal-time commutators for two local operators, $V_1(\sigma,\tau)$ and $V_2(\sigma,\tau)$, are obtained as the limit

$$\left[V_1(\sigma,\tau), V_2(\sigma',\tau)\right] = \lim_{\epsilon\to 0}\Big(V_1(\sigma,\tau+\epsilon)V_2(\sigma',\tau) - V_2(\sigma',\tau+\epsilon)V_1(\sigma,\tau)\Big). \tag{22}$$

Using the OPEs (21) with the commutation relation (22), we find

$$\frac{1}{2\pi i}[J^A(\sigma,\tau),\bar{J}^B(\sigma',\tau)] = -f_C^{AB}\,\nu\delta(\sigma-\sigma')\big[J^C(\sigma,\tau) - \bar{J}^C(\sigma,\tau)\big] + \dots, \tag{23a}$$

$$\frac{1}{2\pi i}[J^A(\sigma,\tau),J^B(\sigma',\tau)] = -k\delta^{AB}\delta'(\sigma-\sigma') + f_C^{AB}\delta(\sigma-\sigma')\big[\lambda J^C(\sigma,\tau) + \mu\bar{J}^C(\sigma,\tau)\big] + \dots, \tag{23b}$$

where we have used the following identities for the Dirac delta function,

$$\lim_{\epsilon\to 0}\left(\frac{1}{\sigma-i\epsilon} - \frac{1}{\sigma+i\epsilon}\right) = 2\pi i\delta(\sigma), \tag{24a}$$

$$\lim_{\epsilon\to 0}\left(\frac{\sigma+i\epsilon}{(\sigma-i\epsilon)^2} - \frac{\sigma-i\epsilon}{(\sigma+i\epsilon)^2}\right) = 2\pi i\delta(\sigma), \tag{24b}$$

$$\lim_{\epsilon\to 0}\left(\frac{1}{(\sigma-i\epsilon)^2} - \frac{1}{(\sigma+i\epsilon)^2}\right) = -2\pi i\delta'(\sigma). \tag{24c}$$

Next, we compactify the theory on a cylinder of circumference $2\pi$, that is to say, we impose the constraint $\sigma = \sigma + 2\pi$. With this compactification, the currents admit the mode expansions

$$J(\sigma,\tau) = i\sum_{n\in\mathbb{Z}} e^{-in\sigma} J_n(\tau), \tag{25a}$$

$$\bar{J}(\sigma,\tau) = -i\sum_{n\in\mathbb{Z}} e^{-in\sigma} \bar{J}_n(\tau), \tag{25b}$$

where we have used the identity

$$\delta(\sigma) = \frac{1}{2\pi}\sum_{n\in\mathbb{Z}} e^{in\sigma}. \tag{26}$$

The zero modes being

$$J_0^A = \frac{1}{2\pi i}\int_{-\pi}^{\pi} d\sigma J^A(\sigma,\tau), \tag{27a}$$

$$\bar{J}_0^A = -\frac{1}{2\pi i}\int_{-\pi}^{\pi} d\sigma \bar{J}^A(\sigma,\tau) \tag{27b}$$

(since they are conserved by Noether's theorem, they do not depend on the time $\tau$), we obtain from (23)

$$\left[J_0^A, J_0^B\right] = f_C^{AB}\left(\lambda J_0^C - \mu \bar{J}_0^C\right), \tag{28a}$$

$$\left[J_0^A, \bar{J}_0^B\right] = f_C^{AB}\, \nu\left(J_0^C + \bar{J}_0^C\right), \tag{28b}$$

and finally

$$\left[J_0^A + \bar{J}_0^A, J_0^B + \bar{J}_0^B\right] = i f_C^{AB}(\lambda - \mu + 2\nu)\left(J_0^C + \bar{J}_0^C\right). \tag{29}$$

The sums $J_0^A + \bar{J}_0^A$ are (proportional to) the global $O(n)$ charges in the model. We now decide to fix the global normalization in (21) so that the zero modes exactly satisfy the global $O(n)$ commutation relations.[2] This will make easier the comparison with numerical results, to be discussed later. It follows that we should have

$$\lambda - \mu + 2\nu = 1. \tag{30}$$

However, it also follows from (21) that

$$\langle J^A(z_1)J^B(z_2)J^C(z_3)\rangle = k\lambda \frac{f_C^{AB}}{(z_1-z_2)(z_2-z_3)(z_3-z_1)}, \tag{31a}$$

$$\langle J^A(z_1)J^B(z_2)\bar{J}^C(z_3)\rangle = k\mu \frac{f_C^{AB}(\bar{z}_1-\bar{z}_2)}{(z_1-z_2)^2(\bar{z}_3-\bar{z}_2)(\bar{z}_3-\bar{z}_1)}. \tag{31b}$$

By symmetry, it follows that $\mu = \nu$; alternatively this can also be deduced from the constraint $\bar{\partial}J - \partial\bar{J} = 0$ applied to (21). We are thus left with the condition

$$\lambda + \mu = 1. \tag{32}$$

Recall now that the ratio of three-point functions in (31) also obeys the shift equation (17): solving these two simple equations for $\lambda$ and $\mu$, we find

$$\lambda = \frac{\beta^2 + 1}{2\beta^2}, \quad\text{and}\quad \mu = \frac{\beta^2 - 1}{2\beta^2}. \tag{33}$$

---

[2]We therefore do *not* choose $k = B_J = 1$, as would be natural for "standard" primary fields.

These agree with known examples[3] of $\lambda, \mu$ for specific values of the central charge $c$:

$$
\begin{array}{c|c|c}
\beta^2 & c & (\lambda, \mu) \\
\hline
1 & 1 & (1, 0) \\
2 & -2 & \left(\frac{3}{4}, \frac{1}{4}\right)
\end{array}
\tag{34}
$$

Formulas of the same nature appear in [23], with $\lambda$ and $\mu$ now given by rational functions of the amplitude of the kinetic term and the level in deformed supergroup WZW models.

## 4 The current four-point functions

To proceed, we now consider the current four-point function $\langle JJJJ \rangle$ by using the conformal bootstrap technique. Using the degenerate shift equations (13) and (18) with $\langle JJJJ \rangle$, we can then obtain similar results for other current four-point functions, such as $\langle JJ\bar{J}\bar{J} \rangle$ and $\langle \bar{J}\bar{J}\bar{J}\bar{J} \rangle$. The goal of this section is to compute exactly the leading terms in the current-current OPEs (21a).

### 4.1 Conformal bootstrap

Recall that the current $J(z, \bar{z})$ transforms in the adjoint representation [11], and it carries a label that we have denoted by $A, B, C, \ldots$ so far, corresponding in fact to a pair of $O(n)$ tensor indices $a, b, c \ldots$, which are antisymmetric under permutation. We can regard the $O(n)$ tensor indices as two lines originating from the point $(z, \bar{z})$. Connecting these lines is then equivalent to contracting the indices. For example, we can represent $\langle J^{ab} J^{ab} \rangle$ as the following diagram:

$$
\langle J^{ab} J^{ab} \rangle : \qquad
\tag{35}
$$

Notice that there are some subtleties in the above construction, having to do with anti-symmetrization, that are discussed in more detail in Appendix A. For the current four-point function, there are 6 different inequivalent ways of contracting the indices, which gives rise to 6 different diagrams:

$$
\tag{36}
$$

where $K_i$ and $L_i$ are simply the name for each diagram. The diagrams in (36) are examples of *combinatorial maps*, two-dimensional graphs which parameterize correlation functions in loop models [24]. The two different sets of objects in (36) and (9) are just different choices of bases for crossing-symmetric solutions of the current four-point function: these two bases are related by a linear transformation, which will be discussed in detail in Appendix A.

---

[3]See below for $\beta^2 = 2$.

Let us now decompose the current four-point function in the base (36). The $s$-channel decomposition reads

$$\langle J(z,\bar{z})J(0)J(\infty)J(1)\rangle = \sum_{\Lambda\in\{K_i,L_i\}} \Lambda F^{\Lambda}_{(s)}(z,\bar{z}),\tag{37}$$

where we have used the global conformal invariance to fix the positions $\{z_2, z_3, z_4\}$ to be $\{0, \infty, 1\}$, and $F^{\Lambda}_{(s)}$ are crossing-symmetry solutions, which depend on the positions, the conformal dimensions, and the central charge. These crossing-symmetry solutions can be further decomposed into the interchiral conformal blocks [25], objects which can be completely determined by conformal symmetry and the existence of degenerate fields. The decomposition reads

$$F^{\Lambda}_{(s)}(z,\bar{z}) = \sum_{(r,s)\in\mathcal{S}^{\Lambda}} D^{\Lambda}_{(r,s)} G^{(s)}_{(r,s)}(z,\bar{z}),\tag{38}$$

where $G^{(s)}_{(r,s)}$ are the interchiral conformal blocks, and $D^{\Lambda}_{(r,s)}$ are four-point structure constants. Details on $G^{(s)}_{(r,s)}$ and $D^{\Lambda}_{(r,s)}$ will be discussed below. Moreover, we use $\mathcal{S}^{\Lambda}$ to denote the spectra for the decomposition (38). In particular, the sets $\mathcal{S}^{\Lambda}$ depend on the diagrams $\Lambda$ and have been completely determined in [8, 24]. We summarize them here in table form:

| $\Lambda$ | Spectra | | |
|---|---|---|---|
| | $s$-channel | $t$-channel | $u$-channel |
| $K_1$ | $\mathcal{S}^1_{\text{even}} \cup \{V^D_{\langle 1,1\rangle}\} - \{V_{(3,\pm\frac{2}{3})}\}$ | $\mathcal{S}^2$ | $\mathcal{S}^2$ |
| $K_2$ | $\mathcal{S}^2$ | $\mathcal{S}^1_{\text{even}} \cup \{V^D_{\langle 1,1\rangle}\} - \{V_{(3,\pm\frac{2}{3})}\}$ | $\mathcal{S}^2$ |
| $K_3$ | $\mathcal{S}^2$ | $\mathcal{S}^2$ | $\mathcal{S}^1_{\text{even}} \cup \{V^D_{\langle 1,1\rangle}\} - \{V_{(3,\pm\frac{2}{3})}\}$ |
| $L_1$ | $\mathcal{S}^1$ | $\mathcal{S}^1$ | $\mathcal{S}^2_{\text{even}}$ |
| $L_2$ | $\mathcal{S}^1$ | $\mathcal{S}^2_{\text{even}}$ | $\mathcal{S}^1$ |
| $L_3$ | $\mathcal{S}^2_{\text{even}}$ | $\mathcal{S}^1$ | $\mathcal{S}^1$ |

$$\tag{39}$$

where

$$\mathcal{S}^{\ell} = \{V_{(r,s)} \in (\mathbb{N}+\ell)\times(-1,1] | rs \in \mathbb{Z}\},\tag{40a}$$
$$\mathcal{S}_{\text{even}} = \{V_{(r,s)} \in \mathcal{S} | rs \in 2\mathbb{Z}\},\tag{40b}$$

and the notations for primary fields have been recalled in Section 2.

### 4.1.1 Interchiral conformal blocks

The existence of the degenerate fields $V^D_{\langle 1,s\rangle}$ in the spectrum of the model allows us to analytically determine the ratio of structure constants between any pair of primary fields which have the same first Kac indices, but whose second Kac indices differ by even integers [26]. Using this type of analytic ratios, we can combine a tower of infinitely many Virasoro-conformal blocks into a single object: an *interchiral conformal block* [25]. Schematically, we have

$$G^{(s)}_{(r,s)} = \sum_{j\in 2\mathbb{Z}} \frac{D^{\Lambda}_{(r,s+j)}}{D^{\Lambda}_{(r,s)}} \begin{array}{c} J(z,\bar{z}) \quad J(1) \\ \diagdown \kern-0.5em \diagup \\ V_{(r,s+j)} \\ \diagup \kern-0.5em \diagdown \\ J(0) \quad J(\infty) \end{array},\tag{41}$$

where each diagram represents Virasoro-conformal blocks of $V_{(r,s+j)}$, and all ratios of structure constants in (41) have been determined in [8] (for more detail, see Section 3.1 of that paper).

### 4.1.2 Structure constants

Using the main result of a companion paper [27], we can now display explicitly the formula for the $s$-channel four-point structure constants $D^\Lambda_{(r,s)}$,

$$
D^\Lambda_{(r,s)} = \mathcal{N}^{\delta_{\Lambda,L_i}} \left( q^\Lambda_{(r,s)}(n) - \delta_{\Lambda,K_1} \delta_{(r,s)\in\mathbb{N}^*\times\mathbb{Z}} \frac{2(-1)^{(r+1)s} p_{(r,s)}}{n+(-1)^{r+1} x_r(n)} \right) \frac{C^{\text{ref}}_{JJV_{(r,s)}} C^{\text{ref}}_{V_{(r,s)}JJ}}{B^{\text{ref}}_{V_{(r,s)}}} , \tag{42}
$$

which also includes the structure constant of the identity field $V^D_{\langle 1,1\rangle}$ as follows:

$$
D^\Lambda_{\langle 1,1\rangle^D} = D^\Lambda_{(0,1-\beta^2)} . \tag{43}
$$

As shown by the prefactor in (42), the structure constants of the solutions $F^{L_i}_{(s)}$ in (37) have an extra factor of $\mathcal{N}$, which is thus the relative normalization between the two types of diagrams: $K_i$ and $L_i$. How to fix $\mathcal{N}$ will be discussed at the end of this section. Let us also stress here that the analytic formula (42) is only valid for non-rational $\beta^2$, since structure constants and conformal blocks for rational $\beta^2$ may diverge—for instance, observe that (48) may have pole for rational value of $\beta^2$. While we do not know yet the complete mechanism of how divergences would cancel, we expect that those divergences should always cancel, since four-point functions of the lattice $O(n)$ loop model exist for generic $n$ (including rational $\beta^2$). That is to say, four-point functions of the $O(n)$ CFT are expected to be finite for rational $\beta^2$, provided that the inequality in (1) holds.

The formula (42) is made of four ingredients: $q^\Lambda_{(r,s)}(n)$, the pole term, $C^{\text{ref}}_{JJV_{(r,s)}}$, and $B^{\text{ref}}_{V_{(r,s)}}$. We now explain each of them in detail.

Let us start with the second term inside the parentheses in (42), which we call the *pole term*. The functions $x_r(n)$ in the denominator are polynomials in $n$ which obey the recursion:

$$
n x_r(n) = x_{r-1}(n) + x_{r+1}(n), \quad \text{with} \quad x_1(n) = n, \quad \text{and} \quad x_0(n) = 2 . \tag{44}
$$

For example,

$$
x_2(n) = n^2 - 2 , \tag{45a}
$$
$$
x_3(n) = n(n^2 - 3) , \tag{45b}
$$
$$
x_4(n) = n^4 - 4n^2 + 2 , \tag{45c}
$$
$$
x_5(n) = n(n^4 - 5n^2 + 5) . \tag{45d}
$$

Whenever $(r,s) \in \mathbb{N}^* \times \mathbb{Z}$, the structure constants $D^{K_1}_{(r,s)}$ pick up the pole term in (42), with residue given by

$$
p_{(r,s)} = \prod_{\epsilon=0,2} \prod_{j=-\frac{r-1-\epsilon}{2}}^{\frac{r-1-\epsilon}{2}} 4\cos^2 \pi \left( j\beta^2 + \tfrac{s-\epsilon}{2} \right) . \tag{46}
$$

(The index $j$ may be fractional, but as shown by the notation it is incremented in steps of 1 inside the product.) More precisely, the pole term appears only in the structure constants of channels containing the identity field $V^D_{\langle 1,1\rangle}$, so by (39) this only affects the diagram $K_1$ in the $s$-channel expansion, explaining the factor $\delta_{\Lambda,K_1}$ in (42). Still by (39), we would find the same pole term with the same residue in the solutions $F^{K_2}_{(t)}$ and $F^{K_3}_{(u)}$ for the other two channels.

Furthermore, for $r \in 2\mathbb{N}^* + 1$, we find $p_{(r,1)} = 0$ for $r \in 2\mathbb{N}^* + 1$, since (46) always produces a factor $\sin \pi = 0$. Having these vanishing residues is also consistent with the permutation symmetry of (37), which requires any structure constants with $rs \in 2\mathbb{N}^* + 1$ to vanish. Now, recall the relation $\cos(j \arccos \frac{n}{2}) = T_j(\frac{n}{2})$ where $T_j(x)$ are Chebyshev polynomials of the first kind. Therefore, using (1), we can always rewrite the product (46) as polynomials in $n$. For the sake of easy reference, we here display the non-vanishing $p_{r,s}$ for $r \le 5$:

$$p_{(1,0)} = 2\,, \tag{47a}$$

$$p_{(2,0)} = \frac{1}{2}(n-2)^2\,, \tag{47b}$$

$$p_{(2,1)} = \frac{1}{2}(n+2)^2\,, \tag{47c}$$

$$p_{(3,0)} = 8n^4\,, \tag{47d}$$

$$p_{(4,0)} = \frac{1}{2}(n-2)^6(n+1)^4\,, \tag{47e}$$

$$p_{(4,1)} = \frac{1}{2}(n+2)^6(n-1)^4\,, \tag{47f}$$

$$p_{(4,0)} = \frac{1}{2}(n-2)^6(n+1)^4\,, \tag{47g}$$

$$p_{(5,0)} = 8n^8\left(n^2-2\right)^4\,. \tag{47h}$$

The functions $C^{\text{ref}}_{V_{(r_1,s_1)}V_{(r_2,s_2)}V_{(r_3,s_3)}}$ and $B^{\text{ref}}_{V_{(r,s)}}$ in (42) are crucial building blocks of four-point structure constants in the $O(n)$ CFT, and we call them *reference structure constants*. Reference structure constants are universal factors of structure constants models that are independent of the model's global symmetry, namely $O(n)$ symmetry for our case. In other words, reference structure constants serve as references for structure constants of primary fields with the same dimensions but transform in different $O(n)$ representations. These reference structure constants depend only on the conformal dimensions and the central charge, and have the expressions:

$$C^{\text{ref}}_{V_{(r_1,s_1)}V_{(r_2,s_2)}V_{(r_3,s_3)}} = \prod_{\epsilon_1,\epsilon_2,\epsilon_3=\pm} \Gamma_\beta^{-1}\left(\frac{\beta+\beta^{-1}}{2} + \frac{\beta}{2}\left|\sum_i \epsilon_i r_i\right| + \frac{\beta^{-1}}{2}\sum_i \epsilon_i s_i\right)\,, \tag{48a}$$

$$B^{\text{ref}}_{V_{(r,s)}} = \frac{(-1)^{rs}}{2\sin\left(\pi(\text{frac}(r)+s)\right)\sin\left(\pi(r+\beta^{-2}s)\right)}\prod_{\pm,\pm}\Gamma_\beta^{-1}\left(\beta \pm \beta r \pm \beta^{-1}s\right)\,, \tag{48b}$$

where we use $\text{frac}(r) := r - \lfloor r \rfloor$ to denote the fractional part of $r \in \frac{1}{2}\mathbb{N}^*$, for example $\text{frac}(2) = 0$ whereas $\text{frac}(\frac{3}{2}) = \frac{1}{2}$. The functions $\Gamma_\beta(x)$ appearing in (48) are Barnes' double Gamma functions, which obey the functional relations (shift equations)

$$\frac{\Gamma_\beta(x+\beta)}{\Gamma_\beta(x)} = \sqrt{2\pi}\beta^{\beta x - \frac{1}{2}}\Gamma^{-1}(\beta x)\,, \tag{49a}$$

$$\frac{\Gamma_\beta(x+\beta^{-1})}{\Gamma_\beta(x)} = \sqrt{2\pi}\beta^{\frac{1}{2}-\beta^{-1}x}\Gamma^{-1}(\beta^{-1}x)\,. \tag{49b}$$

Notice that $B^{\text{ref}}_{V_{(r,s)}}$ is not well defined for $r,s \in \mathbb{N}^* \times \mathbb{Z}$ due to the poles of double Barnes' Gamma functions. For this case, $B^{\text{ref}}_{V_{(r,s)}}$ can be computed by taking the limit from generic values of $s$. Moreover, observe that the special case $C^{\text{ref}}_{V_{(0,s_1)}V_{(0,s_2)}V_{(0,s_3)}}$ coincides with the $c \le 1$ Liouville structure constant of [28].

As to the final ingredient of (42), the functions $q^\Lambda_{(r,s)}(n)$ are polynomials in $n$, which depend on the diagram $\Lambda$. They can be uniquely determined by the crossing-symmetry equation.

Guided by the results obtained in [27] for $q^{\Lambda}_{(r,s)}(n)$ with $r \leq 5$, we conjecture that the degrees of $q^{\Lambda}_{(r,s)}$ for any $r,s$ in the $s$-channel obey the bounds:

$$\deg q^{K_1}_{(r,s)} \leq r^2 - 4, \quad \deg q^{K_2,K_3}_{(r,s)} \leq r^2 - 2, \quad \text{and} \quad \deg q^{L_1,L_2,L_3}_{(r,s)} \leq r(r-1). \tag{50}$$

Observe from (4.1.2) that the degrees of $q^{\Lambda}_{(r,s)}$ could increase quickly as we increase $r$, therefore computing $q^{\Lambda}_{(r,s)}$ for higher $r$ could require bootstrapping the four-point functions (37) for many values of $n$. For instance, if we want $q^{\Lambda}_{(r,s)}$ for $r = 6$, we would need to compute the diagrams $K_2$ for 34 different value of $n$: this is difficult to reach by our standard laptop. This is why we only computed the examples of $q^{\Lambda}_{(r,s)}$ for $r \leq 5$. We do not know, at this stage, an analytical means of obtaining the general expression for coefficients of $q^{\Lambda}_{(r,s)}$. Nonetheless, our numerical bootstrap results are accurate enough to determine exactly several examples of $q^{\Lambda}_{(r,s)}$. Furthermore, the degenerate fields put constraints on the polynomials $q^{\Lambda}_{(r,s)}$ as follows:

$$q^{\Lambda}_{(r,s)}\Big|_{\langle JJJJ\rangle} = q^{\Lambda}_{(r,s)}\Big|_{\langle \bar{J}\bar{J}\bar{J}\bar{J}\rangle} = q^{\Lambda}_{(r,s)}\Big|_{\langle \bar{J}\bar{J}JJ\rangle}. \tag{51}$$

To arrive at the above relation, we first recall the analytic expressions of the residues $p_{r,s}$ in (42) for the four-point functions $\langle JJJJ\rangle$, $\langle \bar{J}\bar{J}\bar{J}\bar{J}\rangle$, and $\langle \bar{J}\bar{J}JJ\rangle$ from Section 2 of the companion paper [27]. We find

$$p_{(r,s)}\Big|_{\langle JJJJ\rangle} = p_{(r,s)}\Big|_{\langle \bar{J}\bar{J}\bar{J}\bar{J}\rangle} = p_{(r,s)}\Big|_{\langle \bar{J}\bar{J}JJ\rangle}. \tag{52}$$

Then, we recall that the ratio between $D_{(r,s)}\Big|_{\langle JJJJ\rangle}$ and $D_{(r,s)}\Big|_{\langle \bar{J}\bar{J}\bar{J}\bar{J}\rangle}$ is completely fixed by the degenerate-shift equation (13). Next, using (48), it can be shown that $D^{\text{ref}}_{(r,s)}\Big|_{\langle JJJJ\rangle}$ and $D^{\text{ref}}_{(r,s)}\Big|_{\langle \bar{J}\bar{J}\bar{J}\bar{J}\rangle}$ coincide with such degenerate-shift equation. Together with (52), we can write

$$\frac{D_{(r,s)}\Big|_{\langle JJJJ\rangle}}{D_{(r,s)}\Big|_{\langle \bar{J}\bar{J}\bar{J}\bar{J}\rangle}} = \frac{D^{\text{ref}}_{(r,s)}\Big|_{\langle JJJJ\rangle}}{D^{\text{ref}}_{(r,s)}\Big|_{\langle JJJJ\rangle}} \implies q^{\Lambda}_{(r,s)}\Big|_{\langle JJJJ\rangle} = q^{\Lambda}_{(r,s)}\Big|_{\langle \bar{J}\bar{J}\bar{J}\bar{J}\rangle}, \tag{53a}$$

$$\frac{D_{(r,s)}\Big|_{\langle JJJJ\rangle}}{D_{(r,s)}\Big|_{\langle \bar{J}\bar{J}\bar{J}\bar{J}\rangle}} = \frac{D^{\text{ref}}_{(r,s)}\Big|_{\langle JJJJ\rangle}}{D^{\text{ref}}_{(r,s)}\Big|_{\langle \bar{J}\bar{J}JJ\rangle}} \implies q^{\Lambda}_{(r,s)}\Big|_{\langle \bar{J}\bar{J}\bar{J}\bar{J}\rangle} = q^{\Lambda}_{(r,s)}\Big|_{\langle \bar{J}\bar{J}JJ\rangle}. \tag{53b}$$

Therefore, we will only display explicitly the polynomials $q^{\Lambda}_{(r,s)}$ for $\langle JJJJ\rangle$. From [27], the polynomials $q^{\Lambda}_{(r,s)}$ obey the relations: $q^{\Lambda}_{(r,s)} = q^{\Lambda}_{(r,-s)}$ and $q^{\Lambda}_{(r,s)} = q^{\Lambda}_{(r,s+2)}$. Taking into account these symmetries, let us now display the independent polynomials for $r \leq 3$ (or 4 in some cases). We refrain from displaying the polynomials at $r = 5$ because they have very complicated expressions due to their high degrees. For the $s$-channel of $K_1$, we find

$$q^{K_1}_{\langle 1,1\rangle^D} = 1, \tag{54a}$$

$$2q^{K_1}_{(4,0)} = n^2 \left(n^2 - 4\right)^3 \left(n^2 - 1\right)^2, \tag{54b}$$

$$2q^{K_1}_{(4,\frac{1}{2})} = -n^4 (n^2 - 3)^2 (n^2 - 2)^2, \tag{54c}$$

$$2q^{K_1}_{(4,1)} = n^2 \left(n^2 - 4\right)^3 \left(n^2 - 1\right)^2. \tag{54d}$$

For the $s$-channel of $K_2$,

$$2q_{(2,0)}^{K_2} = n^2 - 4, \tag{55a}$$

$$2q_{(2,\frac{1}{2})}^{K_2} = -n^2, \tag{55b}$$

$$2q_{(2,1)}^{K_2} = n^2 - 4, \tag{55c}$$

$$3q_{(3,0)}^{K_2} = n^3(n^2 - 4)(n^2 - 12), \tag{55d}$$

$$3q_{(3,\frac{1}{3})}^{K_2} = -n(n^2 - 9)(n^2 - 1)^2, \tag{55e}$$

$$3q_{(3,\frac{2}{3})}^{K_2} = n(n^2 - 3)^2(n^2 - 1), \tag{55f}$$

$$3q_{(3,1)}^{K_2} = -n^3(n^2 - 4)^2. \tag{55g}$$

For the $s$-channel of $L_2$,

$$q_{(2,0)}^{L_2} = n^2 - 4, \tag{56a}$$

$$q_{(2,1)}^{L_2} = -(n^2 - 4), \tag{56b}$$

$$3q_{(3,0)}^{L_2} = 32n^2(n^2 - 4), \tag{56c}$$

$$3q_{(3,\frac{2}{3})}^{L_2} = -4(n^2 - 1)(n^2 - 3), \tag{56d}$$

$$q_{(4,0)}^{L_2} = (n-2)^3(n+1)^2(n+2)\left[n^6 + 2n^5 - 2n^4 - 8n^3 + 9n^2 - 4n + 4\right], \tag{56e}$$

$$q_{(4,\frac{1}{2})}^{L_2} = 2n^5(n^2 - 3)(n^2 - 2), \tag{56f}$$

$$q_{(4,1)}^{L_2} = -(n-2)(n-1)^2(n+2)^3\left[n^6 - 2n^5 - 2n^4 + 8n^3 + 9n^2 + 4n + 4\right]. \tag{56g}$$

And finally, for the $s$-channel of $L_1$,

$$q_{(1,0)}^{L_1} = 1, \tag{57a}$$

$$q_{(1,1)}^{L_1} = 1, \tag{57b}$$

$$q_{(2,0)}^{L_1} = -(n-2), \tag{57c}$$

$$q_{(2,\frac{1}{2})}^{L_1} = -n, \tag{57d}$$

$$q_{(2,1)}^{L_1} = -(n+2), \tag{57e}$$

$$3q_{(3,0)}^{L_1} = 2n^2(n^4 - 6n^2 + 32), \tag{57f}$$

$$3q_{(3,\frac{1}{3})}^{L_1} = (n^2 - 1)(n^4 - n^2 + 18), \tag{57g}$$

$$3q_{(3,\frac{2}{3})}^{L_1} = -(n^2 - 1)(n^2 - 2)(n^2 - 3), \tag{57h}$$

$$3q_{(3,1)}^{L_1} = -2n^2(n^2 - 4)^2. \tag{57i}$$

In any CFT, four-point structure constants can be factorized into products of three-point structure constants, and it is clear that the reference structure constants in (42) admits such factorizations, for instance we expect

$$C_{JJV} \sim (\text{polynomial in } n) \times C_{JJV}^{\text{ref}}, \tag{58}$$

where the above polynomial in $n$ would result from rewriting $q_{(r,s)}^\Lambda$ as a sum over products of three-point functions. However it is not yet clear to us how to rewrite the polynomials $q_{(r,s)}^\Lambda$ in

terms of three-point functions. For instance, $q^{L_2}_{(4,1)}$ cannot be factorized, but it could be a sum of factorized polynomials. This would suggest that fusion rules in the $O(n)$ CFT come with non-trivial multiplicities, which also seems consistent with the facts that primary fields in (5) have non-trivial multiplicities. We leave this issue of factorization for the future work.

Furthermore, we have argued that the polynomials $q^{\Lambda}_{(r,s)}$ can be uniquely determined by the crossing-symmetry equation. Admittedly we do not yet know how to derive them analytically, and our argument is based on purely numerical observations. Having their analytic derivation would mean having a proof for the crossing symmetry of the $O(n)$ CFT. This issue is beyond the scope of the present paper, but we hope to return to it in the future.

## 4.2 Numerical results

Let us now compare the exact formula (42) with numerical results for the same structure constants obtained by the numerical bootstrap of [8]. The code for numerical results in this paper can be found in the notebook `On_current.ipynb` in [29]. Throughout this section, we shall consider the four-point function $\langle \bar{J}\bar{J}\bar{J}\bar{J}\rangle$, instead of $\langle JJJJ\rangle$. Our bootstrap program in [8] is designed for four-point functions of primary fields whose both Kac indices are non-negative integers. Since the four-point functions $\langle \bar{J}\bar{J}\bar{J}\bar{J}\rangle$ and $\langle JJJJ\rangle$ have the same polynomials $q^{\Lambda}_{(r,s)}$, as discussed in (51), we do not need to adjust our bootstrap program and can just use the results for $\langle \bar{J}\bar{J}\bar{J}\bar{J}\rangle$. For convenience, we first summarize the main ideas of the numerical bootstrap approach of [8].

1. We truncate the infinite spectra $\mathcal{S}^{\Lambda}$ of (38) by introducing a cutoff $\Delta_{\max}$ on conformal dimensions in $\mathcal{S}^{\Lambda}$, including the descendant fields, so that only fields with $\Re(\Delta + \bar{\Delta}) \leq \Delta_{\max}$ are included in the sum. This truncation applies, in particular, to the sums (38) and (41), and to the conformal blocks in (41).

2. Therefore, the crossing-symmetry equation of the truncated four-point function is a linear system for the structure constants. For instance, the crossing-symmetry equation of the truncated diagram $K_1$ in (36) reads

$$\sum_{\substack{(r,s)\in\mathcal{S}^{K_1}}}^{\Re(\Delta+\bar{\Delta})\leq\Delta_{\max}} D^{K_1}_{(r,s)} G^{(s)}_{(r,s)}(z,\bar{z}) = \sum_{\substack{(r,s)\in\mathcal{S}^{K_2}}}^{\Re(\Delta+\bar{\Delta})\leq\Delta_{\max}} D^{K_2}_{(r,s)} G^{(t)}_{(r,s)}(z,\bar{z}) = \sum_{\substack{(r,s)\in\mathcal{S}^{K_3}}}^{\Re(\Delta+\bar{\Delta})\leq\Delta_{\max}} D^{K_3}_{(r,s)} G^{(u)}_{(r,s)}(z,\bar{z}),$$
(59)

where the interchiral blocks are completely known and the structure constants are the unknowns. Furthermore, using the permutation symmetry, we find that the $t$- and $u$-channel structure constants of $K_1$ are equal to the $s$-channel structure constants of $K_2$ and $K_3$, respectively.

3. In practice, we can obtain a linear system out of the crossing-symmetry equation (59) by computing (59) at as many positions $\{z, \bar{z}\}$ as the number of unknown structure constants. Solving this linear system gives us numerical values for the structure constants, whose numerical errors are controlled by the cutoff $\Delta_{\max}$: the data becomes more accurate as we increase $\Delta_{\max}$.

For instance, let us compute the ratio between the $s$-channel structure constants $D^{K_1}_{(1,1)^D}$ and $D^{K_1}_{(2,0)}$, by deploying the the numerical bootstrap at a generic value $\beta = 0.8 + 0.1i$ and using

various cutoffs $\Delta_{\max}$:

| $\Delta_{\max}$ | $\Re\left(D^{K_1}_{\langle 1,1\rangle^D}/D^{K_1}_{(2,0)}\right)$ |
|---|---|
| 20 | $\underline{-19.4815718559603581918739888296}35529835459930433020$ |
| 40 | $\underline{-19.48158881201383856106449024045880}763822971041937$ |
| 60 | $\underline{-19.48158881201383855422912679022}375189584131387574$ |

$$(60)$$

On the other hand, using the exact formula (42), we find

$$
\frac{D^{K_1}_{\langle 1,1\rangle^D}}{D^{K_1}_{(2,0)}} = \frac{n+1}{n-2}
$$

$$
\times \frac{\beta^{-\beta^2-1}\Gamma_\beta\left(\frac{2}{\beta}-\beta\right)\Gamma_\beta^8\left(\frac{3\beta}{2}+\frac{1}{2\beta}\right)\Gamma_\beta^2\left(\frac{5\beta}{2}-\frac{1}{2\beta}\right)\Gamma_\beta^2\left(\frac{5\beta}{2}+\frac{3}{2\beta}\right)\Gamma_\beta^2\left(\frac{\beta^2-1}{2\beta}\right)\Gamma_\beta^2\left(\frac{\beta^2+3}{2\beta}\right)}{\sin\left(2\pi\beta^2\right)\Gamma_\beta\left(\frac{1}{\beta}-2\beta\right)\Gamma_\beta^3\left(\frac{1}{\beta}\right)\Gamma_\beta^6(\beta)\Gamma_\beta(3\beta)\Gamma_\beta^2\left(\beta+\frac{2}{\beta}\right)\Gamma_\beta\left(2\beta-\frac{1}{\beta}\right)\Gamma_\beta^3\left(2\beta+\frac{1}{\beta}\right)\Gamma\left(-\beta^2\right)},
$$

$$(61)$$

where the rational function in $n$ on the first line comes from the residue factor in (42) (recall that $q^{K_1}_{(2,0)}=0$), while the combination of double Gamma functions on the second line comes from the reference structure constants. To compare the analytic result (61) with (60), we compute (61) numerically,

$$
\Re\left(\frac{D^{K_1}_{\langle 1,1\rangle^D}}{D^{K_1}_{(2,0)}}\bigg|_{\beta=0.8+0.1i}\right) = -19.48158881201383855422912679022255620616\cdots. \qquad (62)
$$

In the table (60), we have underlined the decimals of the numerical results that coincide with (62). It is seen that the numerical result with $\Delta_{\max}=60$ in (60) agrees with (62) to a precision of 32 significant digits. Such excellent agreement strongly confirms that (42) is indeed correct.

For other structure constants, let us first point out that it is sufficient to display the results for the $s$-channel expansions of the diagrams $K_1$, $K_3$, $L_1$ and $L_2$, since the $s$-channel expansion of $K_2$ and $L_3$ in (36) can be obtained by applying a permutation of the points $\{z_3,\bar{z}_3\}$ and $\{z_4,\bar{z}_4\}$ to $K_3$ and $L_1$, respectively. In particular, we have the relations:

$$
D^{K_2}_{(r,s)} = (-1)^{rs} D^{K_3}_{(r,s)}, \qquad (63a)
$$

$$
D^{L_1}_{(r,s)} = (-1)^{rs} D^{L_3}_{(r,s)}. \qquad (63b)
$$

Below, we display the numerical data for the structure constants with $r \le 4$ of $K_1$, $K_3$, $L_1$, and $L_2$, and we choose the cutoff $\Delta_{\max}=40$, still with the parameter value $\beta=0.8+0.1i$. Furthermore, we normalize $K_1$ and $L_2$ such that $D^{s\text{-channel}}_{(2,0)}=1$, whereas $K_3$ and $L_1$ are normalized such that the $u$-channel structure constant $D^{u\text{-channel}}_{(2,0)}$ is 1. Furthermore, we have set the $s$-channel structure constant $D^{K_1}_{(3,\pm\frac{2}{3})}$ to be zero, according to (39). With this choice of parameters, results from the numerical bootstrap coincide with the exact formula (42) to a precision of around 15–20 digits.

The $s$-channel expansion of $K_1$ at $\beta = 0.8 + 0.1i$ and $\Delta_{\max} = 40$.

| $(r,s)$ | $\Re$(Numerical bootstrap) | $\Re$(Exact formula/$D_{(2,0)}^{s\text{-channel}}$) |
|---|---|---|
| $\langle 1,1 \rangle^D$ | $-19.48158881201384$ | $-19.481588812013838554$ |
| $(1,0)$ | $8.02892221860604$ | $8.0289222186060403289$ |
| $(2,1)$ | $-15.862593142515996$ | $-15.862593142515996439$ |
| $(3,0)$ | $-0.08984306338246206$ | $-0.089843063382462066314$ |
| $(3,\pm\frac{2}{3})$ | $0$ | $0$ |
| $(4,0)$ | $0.009980294352557147$ | $0.0099802943525571467673$ |
| $(4,\frac{1}{2})$ | $-0.022950732471538973$ | $-0.022950732471538973019$ |
| $(4,-\frac{1}{2})$ | $0.0008112374680706194$ | $0.00081123746807061936589$ |
| $(4,1)$ | $0.024769209718927935$ | $0.024769209718927933329$ |

(64)

The $s$-channel expansion of $K_3$ at $\beta = 0.8 + 0.1i$ and $\Delta_{\max} = 40$.

| $(r,s)$ | $\Re$(Numerical bootstrap) | $\Re$(Exact formula/$D_{(2,0)}^{u\text{-channel}}$) |
|---|---|---|
| $(2,0)$ | $2.2845580175551414$ | $2.2845580175551413865$ |
| $(2,\frac{1}{2})$ | $-2.61053147059997$ | $-2.6105314705999700449$ |
| $(2,-\frac{1}{2})$ | $-0.845307560519607$ | $-0.84530756050649992493$ |
| $(2,1)$ | $-9.999899257657525$ | $-9.9998992576575245983$ |
| $(3,0)$ | $-0.6259773425430574$ | $-0.62597734254305740848$ |
| $(3,\frac{1}{3})$ | $-1.598818949423807$ | $-1.5988189494238070347$ |
| $(3,-\frac{1}{3})$ | $0.06275194007325292$ | $0.062751940073252918708$ |
| $(3,\frac{2}{3})$ | $-1.9359217996470641$ | $-1.9359217996470640473$ |
| $(3,-\frac{2}{3})$ | $0.0172246923663488$ | $0.01722469236634879955$ |
| $(3,1)$ | $-0.03658002773884726$ | $-0.036580027738847262748$ |

(65)

The $s$-channel expansion of $L_2$ at $\beta = 0.8 + 0.1i$ and $\Delta_{\max} = 40$.

| $(r,s)$ | $\Re$(Numerical bootstrap) | $\Re$(Exact formula/$D_{(2,0)}^{s\text{-channel}}$) |
|---|---|---|
| $(2,1)$ | $-2.361190498900769$ | $-2.3611904989007686127$ |
| $(3,0)$ | $-0.00169484795113024$ | $-0.0016948479511302399079$ |
| $(3,\frac{2}{3})$ | $0.3210069692438823$ | $0.32100696924388227955$ |
| $(3,-\frac{2}{3})$ | $0.004871812705290773$ | $0.0048718127052907728725$ |
| $(4,0)$ | $0.0013425212052786142$ | $0.0013425212052786140583$ |
| $(4,\frac{1}{2})$ | $0.003341488652166565$ | $0.0033414886521665653114$ |
| $(4,-\frac{1}{2})$ | $-0.000025014269611482264$ | $-0.000025014269611482263912$ |
| $(4,1)$ | $-0.004092010932285123$ | $-0.0040920109322851230015$ |

(66)

The $s$-channel expansion of $L_1$ at $\beta = 0.8 + 0.1i$ and $\Delta_{\max} = 40$.

| $(r,s)$ | $\Re(\text{Numerical bootstrap})$ | $\Re(\text{Exact formula}/D^{u\text{-channel}}_{(2,0)})$ |
|---|---|---|
| $(1,0)$ | $-0.837\,344\,424\,545\,1527$ | $-0.837\,344\,424\,545\,152\,645\,65$ |
| $(1,1)$ | $-0.440\,124\,757\,133\,4242$ | $-0.440\,124\,757\,133\,424\,181\,9$ |
| $(2,0)$ | $-0.310\,923\,294\,757\,4724$ | $-0.310\,923\,294\,757\,472\,386\,22$ |
| $(2,\frac{1}{2})$ | $0.043\,517\,729\,818\,157\,457$ | $0.043\,517\,729\,818\,157\,457\,078$ |
| $(2,-\frac{1}{2})$ | $0.217\,309\,139\,486\,404\,96$ | $0.217\,309\,139\,483\,146\,103\,66$ |
| $(2,1)$ | $-1.781\,999\,088\,411\,4574$ | $-1.781\,999\,088\,411\,457\,347\,3$ |
| $(3,0)$ | $-0.003\,457\,370\,114\,695\,005$ | $-0.003\,457\,370\,114\,695\,005\,371\,3$ |
| $(3,\frac{1}{3})$ | $0.007\,797\,247\,811\,040\,619$ | $0.007\,797\,247\,811\,040\,614\,080\,8$ |
| $(3,-\frac{1}{3})$ | $0.000\,396\,394\,247\,792\,850\,7$ | $0.000\,396\,394\,247\,792\,853\,003\,7$ |
| $(3,\frac{2}{3})$ | $-0.362\,505\,201\,829\,3606$ | $-0.362\,505\,201\,829\,360\,638\,23$ |
| $(3,-\frac{2}{3})$ | $-0.002\,546\,405\,571\,949\,7786$ | $-0.002\,546\,405\,571\,949\,779\,390\,9$ |
| $(3,1)$ | $0.627\,554\,657\,208\,8655$ | $0.627\,554\,657\,208\,865\,456\,38$ |

$$(67)$$

## 4.3 The level parameter $k$

Let us now translate the structure constants (42) of the current four-point function (37) into the OPE coefficients of the current-current OPE (21a). We are particularly interested in computing the level parameter $k$. From the current-current OPE (21a), we deduce that it is related to the $s$-channel structure constants of (37) as follows:

$$D^{K_1}_{\langle 1,1 \rangle^D} = 16k^2\,, \tag{68a}$$

$$D^{L_1}_J = -D^{L_3}_J = 4\lambda^2 k\,, \tag{68b}$$

$$D^{L_1}_{\bar{J}} = -D^{L_3}_{\bar{J}} = 4\mu^2 k\,, \tag{68c}$$

where the factors of 4 ensure us that $k(n \to 2) = \frac{1}{2}$. Recalling (32) and (33), we find that $k$ is given by

$$k = \frac{1}{4}\left(1 + \frac{\mu}{\lambda}\right)^{-2} \frac{D^{K_1}_{\langle 1,1 \rangle^D}}{D^{L_1}_J}$$

$$= \frac{(\beta^2 + 1)^2}{16\beta^4} \frac{D^{K_1}_{\langle 1,1 \rangle^D}}{D^{L_1}_J}\,. \tag{69}$$

From (42), computing the above ratio of structure constants requires fixing $\mathcal{N}$, which can be done by writing down the relation between diagrams in (36) and $O(n)$ irreducible representations in (9). This is done in Appendix A. To proceed, we rewrite (37) as follows:

$$\langle J(z,\bar{z})J(0)J(\infty)J(1)\rangle = P^{[1111]}F^{[1111]}_{(s)} + P^{[211]}F^{[211]}_{(s)} + P^{[22]}F^{[22]}_{(s)}$$

$$+ P^{[11]}F^{[11]}_{(s)} + P^{[\,]}F^{[\,]}_{(s)} + P^{[2]}F^{[2]}_{(s)}\,, \tag{70}$$

where $P^\lambda$ have an $O(n)$ tensorial structure, to be discussed in detail in the appendix, and $F^\lambda_{(s)}$ are still solutions to the crossing-symmetry equation in the $s$-channel.

The next step is to consider the solution $F_{(s)}^{[1111]}$, in which there are only primary fields that transform as the $O(n)$ representation $[1111]$. From (6), recall that $V_{(2,0)}$ does not transform as $[1111]$ under $O(n)$ symmetry, and therefore, using (39), we find that there is only one possible combination of diagrams in (37) giving such a solution,

$$F_{(s)}^{[1111]} = F_{(s)}^{K_2} + F_{(s)}^{K_3} - 2\frac{D_{(2,0)}^{K_2}}{D_{(2,0)}^{L_3}}F_{(s)}^{L_3} = F_{(s)}^{K_2} + F_{(s)}^{K_3} - \frac{1}{\mathcal{N}}F_{(s)}^{L_3},\tag{71}$$

where we have used (42) to compute explicitly the ratio between $D_{(2,0)}^{K_2}$ and $D_{(2,0)}^{L_3}$. The subtraction in (71) ensures us that the field $V_{(2,0)}$ does not propagate in the $s$-channel of $F^{[1111]}$. On the other hand, the relation between (70) and (37) found in Appendix A yields

$$F_{(s)}^{[1111]} = F_{(s)}^{K_2} + F_{(s)}^{K_3} + 4F_{(s)}^{L_3}.\tag{72}$$

Comparing (71) to the above yields

$$\mathcal{N} = -\frac{1}{4}.\tag{73}$$

Using (73) with (42), we find that

$$\begin{aligned}
\frac{D_{\langle 1,1\rangle^D}^{K_1}}{D_J^{L_1}} &= \frac{1}{\mathcal{N}}\frac{B_J^{\mathrm{ref}}}{B_{V_{\langle 1,1\rangle}^D}^{\mathrm{ref}}}\left(\frac{C_{JJV_{\langle 1,1\rangle}^D}^{\mathrm{ref}}}{C_{JJJ}^{\mathrm{ref}}}\right)^2 \\
&= -\frac{\beta^{\beta^{-2}}\Gamma_\beta\left(\frac{2}{\beta}-\beta\right)\Gamma_\beta^6\left(\frac{1}{\beta}+\beta\right)\Gamma_\beta^2\left(\frac{2}{\beta}+2\beta\right)}{4\sin\left(\frac{\pi}{\beta^2}\right)\Gamma\left(\frac{1}{\beta^2}\right)\Gamma_\beta^4\left(\frac{1}{\beta}\right)\Gamma_\beta^2\left(\frac{2}{\beta}+\beta\right)\Gamma_\beta^3\left(\frac{1}{\beta}+2\beta\right)} \\
&= -8\pi\beta^2\frac{(\beta^2-1)}{(\beta^2+1)^2\sin(\pi\beta^2)}.
\end{aligned}\tag{74}$$

To go from the first to second line, we computed $B_J^{\mathrm{ref}}$ as $\lim_{\epsilon\to 0}B_{(1,-1+\epsilon)}^{\mathrm{ref}}$ because of the poles of double Barnes' Gamma functions in (48): this explains the factor $\sin\left(\frac{\pi}{\beta^2}\right)\Gamma\left(\frac{1}{\beta^2}\right)$. We then employed the functional relations (49) to arrive at the final line. Putting everything together gives us

$$k = -\frac{\pi(\beta^2-1)}{2\beta^2\sin(\pi\beta^2)},\tag{75}$$

where $k$ is as in (21a). How this parameter $k$ corresponds to a "level" in our current algebra, and how this compares with the constant determined in [2], will be discussed in Section 6.

# 5 The limits $n \to \pm 2$

We now discuss two limits in which the $O(n)$ model can be related to free field theories.

## 5.1 The limit $n \to 2$

We have already seen in Section 3 that important simplifications occur when $\beta^2 \to 1$, that is, when $n \to 2$. The level parameter (75) then has the limit

$$k(n\to 2) = \frac{1}{2}.\tag{76}$$

It is not clear, however, what to expect in this limit. From the $O(n)$ point of view, the expected symmetry is just $O(2)$, an Abelian algebra with no structure constants. On the other hand, it is known that the continuum limit of the lattice model at that point can be described as a $Z_2$ orbifold of the $SU(2)_1$ WZW model. In the standard notations, the torus partition function for a free boson, taking values on a circle of radius $R$, is

$$Z_{\text{circ}}(R) = \frac{1}{\eta(q)\bar{\eta}(\bar{q})} \sum_{n,m\in\mathbb{Z}} q^{(m/2R+nR)^2/2} \bar{q}^{(m/2R-nR)^2/2}, \tag{77}$$

where $\eta$ is the Dedekind eta function, and $q$ the modular parameter. We have the following identities between the circle and orbifolded partition functions [30]:

$$Z_{\text{WZW}} = Z_{\text{circ}}(1/\sqrt{2}), \quad \text{and} \quad Z_{\text{orb}}(1/\sqrt{2}) = Z_{\text{circ}}(\sqrt{2}) = Z_{O(n=2)}. \tag{78}$$

Of course, in the orbifolding process the $SU(2)$ symmetry disappears. Introducing the chiral bosonic field $x(z)$ with propagator $\langle x(z)x(w)\rangle = -\ln(z-w)$, the $SU(2)$ currents are obtained via

$$J^{\pm}(z) = e^{\pm i\sqrt{2}x(z)} = J^1 \pm iJ^2, \tag{79a}$$

$$J^3(z) = \frac{i}{\sqrt{2}}\partial x(z), \tag{79b}$$

and obey the purely chiral OPE

$$J^i(z)J^j(w) = \frac{\frac{\delta^{ij}}{2}}{(z-w)^2} + \frac{i\epsilon^{ij}_k}{z-w}J^c(w), \tag{80}$$

where $\epsilon^{ij}_k = \epsilon^{ijk}$ is the totally antisymmetric tensor, $i,j,k \in \{1,2,3\}$. In particular, the $U(1)$ current obeys

$$J^3(z)J^3(w) = \frac{1}{2(z-w)^2}. \tag{81}$$

Comparing with (21a) we thus see that the level is $k = \frac{1}{2}$, in agreement with the limit $n \to 2$ of the general result (75).

While it would seem natural to expect that all but one solution of the bootstrap disappear in the limit $n \to 2$ (since the $O(2)$ model admits only one current), one should remember that the loop model potentially describes more than $O(2)$, even at $n = 2$. In fact, under very simple modifications, the loop model can be re-interpreted [31] in terms of a $U(2)$ model, a fact closely related with the underlying orbifold construction described above (78).

To see what happens to the current four-point function when $n \to 2$, we again use methods from the conformal bootstrap of [8]. For generic $n$, the spectrum (5) gives us 6 solutions to the crossing-symmetry equation, in agreement with the facts that we have 6 fusion channels in (9), or equivalently a basis of 6 combinatorial maps (36). However, as $n \to 2$ our numerical results suggest that we are left with only 3 crossing-symmetric solutions. This can be explained as follows. At $n = 2$, we have the following coincidence of conformal dimensions:

$$\Delta_{(1,-1)} = \Delta_{(1,3)}. \tag{82}$$

Therefore, in addition to the null descendant at level 1, the current $J$ has an extra null descendant at level 3. As mentioned in (78), the partition function of the $O(2)$ CFT is equivalent to the partition function of the $Z_2$ orbifold of the $SU(2)_1$ WZW model. As this model is a unitary CFT, it follows that all null descendants in this model must vanish. In other words, at $c = 1$, the currents are annihilated by the combinations of Virasoro generators,

$$\left(L_{-1}^3 - 4L_{-1}L_{-2} + 6L_{-3}\right)J^{\pm,0} = 0. \tag{83}$$

Therefore, the current four-point functions should be a linear combination of the 3 solutions of the ensuing third-order BPZ differential equation. More explicitly, using (79), we find

$$\langle J^3(z)J^3(0)J^3(\infty)J^3(1)\rangle \propto \frac{1}{z^2} + \frac{1}{(z-1)^2} + 1\,, \tag{84a}$$

$$\langle J^+(z)J^-(0)J^+(\infty)J^-(1)\rangle \propto \frac{1}{z^2(z-1)^2}\,, \tag{84b}$$

$$\langle J^+(z)J^-(0)J^3(\infty)J^3(1)\rangle \propto \frac{1}{z^2} - \frac{2}{z-1}\,. \tag{84c}$$

It is a straightforward exercise to check that each four-point function in (84) is annihilated by the Virasoro generators (83).Nevertheless, we do not yet understand how to associate $J^0$ and $J^\pm$ at the point $n = 2$ to the generic $O(n)$ currents, neither do we know the physical meaning of the other 3 crossing-symmetric solutions that disappear.

Now, comparing (77) with (5), we find that primary fields $V_{(r,s)}$ whose $(r,s) \notin \frac{\mathbb{Z}}{2} \times \frac{\mathbb{Z}}{2}$ are excluded from the partition function (77). One might wonder quite generally what happens to the four-point functions of such fields in the limit $n \to 2$, and what the limit means? While this issue is beyond the scope of this paper, we note that a likely answer is that these four-point functions make sense in a more general supersymmetric theory (see also below) of the type $U(m+2|m)$ with $m > 0$, as proposed in [6]. We hope to discuss this in more detail elsewhere.

## 5.2 The limit $n \to -2$

We now turn to the limit $n \to -2$ in the dilute phase, corresponding to $\beta^2 \to 2$ and the central charge $c = -2$. It is well known that the lattice model underlying the $O(n)$ loop model can be generalized to include fermionic degrees of freedom. The spins then belong to the fundamental representation of a superalgebra of type $OSp(2m+n|2m)$, with $m$ a positive integer. The partition function of such a model with periodic boundary conditions for the fermions in both directions coincides with the one of the $O(n)$ model.[4] This can be used, in particular, to express the limit $n \to -2$ in terms of an $OSP(0|2)$ model, recovering the observation that the loop model at $n = -2$ can be described in terms of symplectic fermions [32] with

$$\langle \eta_1(z,\bar{z})\eta_2(w,\bar{w})\rangle = -\ln|z-w|^2\,. \tag{85}$$

The $OSP(0|2)$ currents then read

$$J^1 = \frac{1}{4}\left(\eta_1\partial\eta_1 + \eta_2\partial\eta_2\right),$$

$$J^2 = \frac{1}{4}\left(\eta_2\partial\eta_2 - \eta_1\partial\eta_1\right),$$

$$J^3 = \frac{1}{4}\left(\eta_1\partial\eta_2 + \eta_2\partial\eta_1\right), \tag{86}$$

together with identical expressions for the $\bar{J}^a$ with $\partial \leftrightarrow -\bar{\partial}$. Observe that these currents obey $\partial\bar{J} = \bar{\partial}J$ (since, we recall, $\partial\bar{\partial}\eta_{1,2} = 0$).

Extending the construction from Section 3 to the orthosymplectic case leads, for $OSP(0|2)$, to the replacement of the tensor $\delta^{AB}$ in the OPEs (21a) with $\eta^{AB} = \eta_{AB} = \text{Diag}(1,-1,-1)$, $A, B, C, D \in \{1, 2, 3\}$, so now we should have formally

$$J^A(z(z,\bar{z})J^B(w,\bar{w}) = \frac{k(n=-2)\eta^{AB}}{(z-w)^2} + \ldots \tag{87}$$

---

[4]The fermions in this construction have integer conformal weights, so the boundary conditions are the same on the cylinder and in the plane.

However, we find, from direct calculation

$$J^A(z,\bar{z})J^B(w,\bar{w}) = \frac{\eta^{AB}}{8(z-w)^2}\left(-1 + \eta_1\eta_2 - \ln|z-w|^2\right) + \dots \tag{88}$$

This can be interpreted by a formal divergence of the anomaly $k \approx -\frac{1}{8}\ln|z-w|^2$, in agreement with the result (75) since, as $n \to -2$, that is $\beta^2 \to 2$, we have $k(n) \to +\infty$.

Further calculations give, e.g.,

$$J^1(z,\bar{z})J^2(w,\bar{w}) = \dots + \frac{3}{4}J^3\frac{1}{z-w} + \frac{1}{4}\bar{J}^2\frac{\bar{z}-\bar{w}}{(z-w)^2} + \dots \tag{89}$$

This corresponds, in the notations of section 3 (and using $f_C^{AB} = -\epsilon^{ABD}\eta_{DC}$), to $\lambda = \frac{3}{4}$, $\mu = \frac{1}{4}$, in agreement with (33), since $\beta^2 = 2$.

As mentioned earlier, counting of the fields with conformal weight $h = \bar{h} = 1$ requires that many of the terms one would normally denote $:J^A\bar{J}^B:$ are in fact zero. This can be illustrated in the case of symplectic fermions, although in this case the number of fields of weight $(1,0)$ is larger than the dimension of the adjoint of $OSp(0|2)$, which gives only three fields. In this theory, there are in fact eight fields (the physical origin of this extra degeneracy is discussed in [33] in terms of spontaneously broken $OSp(1|2)$ symmetry) with weight $(1,0)$: any one of $\partial\eta_1, \partial\eta_2$ multiplied by any one of $1, \eta_1, \eta_2, \eta_1\eta_2$. The fields of weight $(1,1)$ are obtained by multiplying $\partial\eta_i\bar{\partial}\eta_j$ by the same four fields, resulting in a multiplicity of $4 \times 2^2 = 16$, and not $(4 \times 2)^2 = 64$. Obviously, a field such as $:(\eta_1\partial\eta_1)(\eta_1\partial\eta_2): = 0$, etc. As for the currents $J^A$ themselves, since they are all even in fermions, their normal-ordered products can only expand on $\partial\eta_1\bar{\partial}\eta_2$ and $\bar{\partial}\eta_1\partial\eta_2$ multiplied by 1 or $\eta_1\eta_2$, and thus the 9 possible combinations are not all independent.

# 6 Application to loop models

## 6.1 The current-current perturbation of self-avoiding walks

The presence of local currents in the $O(n)$ model suggests the existence of interesting current-current perturbations. In particular, it was proposed in a series of works starting with [1] that an orientation-dependent interaction between neighboring monomers in the self-avoiding walk (SAW) problem—and more generally, in the loop model—should give rise to continuously varying exponents. This followed, in the logic of [1], from the fact that the combination $:J\bar{J}:\big|_{[\,]}$ should be an exactly marginal field with conformal weights $h = \bar{h} = 1$. This point in turn is argued in [1] by analogy with the free bosonic theory underlying the (by now, understood to be incomplete) Coulomb gas construction of the CFT for the loop model.

We have mentioned earlier that, by pure counting, several of the $:J^A\bar{J}^B:$ terms had to vanish. By looking at the OPEs discussed in the previous sections—in particular eqs. (19) and (20)—we see that for $n$ generic (in fact, $n \neq \pm 2$), there are simply *no* such terms. The only fields with weight $(1,1)$ in the loop model are the Jordan partners of the logarithmic field $W_{(1,1)}$. They appear as the bottom and top components of indecomposable modules for the

left or right Virasoro algebra, according to the diagram below:

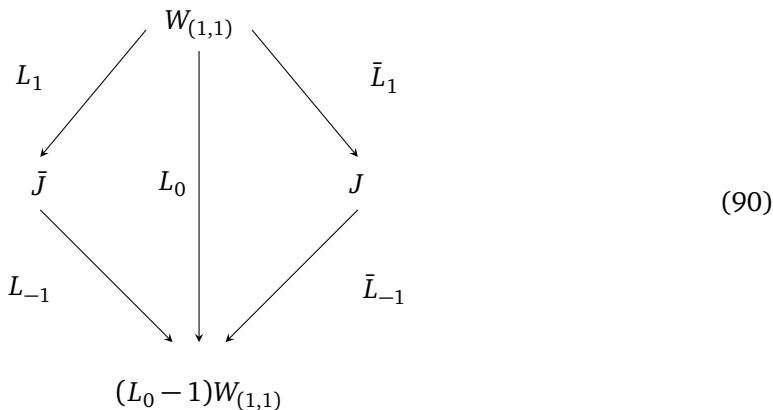

$$(90)$$

The $W_{(1,1)}$ carry labels in the adjoint representation, and thus arise with multiplicity $2 \times d_{[11]}$ in agreement with the counting in the partition function.

It turns out, however, that the fields $W_{(1,1)}$ do *not* appear in normal-ordered expressions such as $:J^A \bar{J}^B:$. On general grounds, we know they might have appeared in OPEs but multiplied by logarithms; whether this means they might contribute a finite part is a moot question, since we have seen earlier—in the discussion below eq. (17)—that they in fact just decouple. Hence we conclude that

$$:J\bar{J}:\Big|_{[\,]} = :J\bar{J}:\Big|_{[11]} = 0. \tag{91}$$

It is thus clear that, in light of a deeper understanding of the loop model CFT, the argument presented in [1] does not work. This should not be too much of a surprise, since the consequences (varying exponents in the presence of orientation-dependent interactions) were never convincingly observed numerically (for discussions, see [10,34,35]).

We can go a bit further and discuss what should in fact be expected from the orientation-dependent interaction considered in [1]. Since the singlet-channel of $J\bar{J}$ exhibits, by (19), on the right-hand side the term

$$J(z,\bar{z})\bar{J}(0)\Big|_{[\,]} \sim \frac{C_{J\bar{J}V_{(2,0)}}}{B_{V_{(2,0)}}} |z|^{2\Delta_{(2,0)}} V_{(2,0)} + \dots, \tag{92}$$

we see that a lattice current-current coupling obtained by bringing two current operators on neighboring sites (the neighboring monomers) will be described, in the continuum limit, by a term where the cut-off still appears explicitly, multiplied by a field of conformal dimensions $(\Delta_{(2,0)}, \Delta_{(2,0)})$. In the dilute case, $\Delta_{(2,0)} > 1$ (for instance $\Delta_{(2,0)} = \frac{35}{24}$ for SAW) and the perturbation is thus irrelevant. In the dense case on the other hand, it is relevant, and coincides in fact with the four-leg crossing perturbation. We thus expect that, while the orientation-dependent interaction of [1] should not change anything to long-distance properties in the dilute $O(n)$ model, it should induce a flow to the $O(n)/O(n-1)$ phase (discussed in [36]) in the dense case.

## 6.2  $O(n)$ currents and lattice loop correlators

We now discuss how to measure correlators of the $O(n)$ CFT on the lattice. We shall assume in what follows that the reader is familiar with both the Hamiltonian and Euclidian versions of these models—see for instance [31] for a recent thorough review.

The abstract generators of the $O(n)$ algebra obey

$$\left[Q^{ab}, Q^{cd}\right] = \delta^{bc}Q^{ad} - \delta^{bd}Q^{ac} - \delta^{ac}Q^{bd} + \delta^{ad}Q^{bc}. \tag{93}$$

In the Hamiltonian version of the lattice model and for the dense case, the space of states is $\mathcal{S}_L^{O(n)} = [1]^L$ for a chain of length $L$. The $O(n)$ symmetry is realized by defining

$$Q^{ab} = \sum_{i=1}^{L} \mathbb{1} \times \ldots \times Q_i^{ab} \times \ldots \times \mathbb{1} , \tag{94}$$

where, in the vector representation, the generator $Q_i$ acts as a matrix with elements

$$(Q^{ab})_{cd} = \delta_c^a \delta_d^b - \delta_d^a \delta_c^b , \tag{95}$$

on the $i^{\text{th}}$ space in the tensor product, and as identity otherwise. In the dilute case, the space of states becomes $\mathcal{S}_L^{O(n)} = ([\,] + [1])^L$, and the action of the generator is similar on $[1]$, while it is trivial on $[\,]$.

Focussing for simplicity on the dense case, the Hamiltonian of the model takes the familiar form

$$H = -\sum e_i , \tag{96}$$

where the $e_i$ are generators of the "unoriented Jones-Temperley-Lieb" algebra $u\mathcal{JTL}$. These generators, for $n$ a positive integer, act by contracting identical colors and projecting onto a pair of identical colors:

$$e_{ab,cd} = \delta_{ab}\delta_{cd} . \tag{97}$$

One can then represent the propagation of a given color as a line, and define the model for $n$ arbitrary using the corresponding formulation of lines and loops.

In this approach, the action of the local generator $Q^{ab}$ corresponds to having a line carrying the color $a$ terminate and be replaced by a new line carrying the color $b$, or the reverse (this time with a "Boltzmann weight" equal to $-1$). While the generators themselves cannot be interpreted for $n$ non-integer, their correlators can, as we shall see below.

To connect with the bootstrap, it is easier to move to the Euclidian version of the model, where an $n$-component spin lives on every vertex of a lattice—we choose the square lattice for convenience—, and the lines and loops are obtained via a high-temperature expansion. This spin becomes the "vector field" of the sigma model in the continuum limit, whose action is

$$A = \frac{1}{2g_\sigma^2} \int dx\,dy \, (\partial_\mu \vec{S})^2 , \tag{98}$$

with the constraint $\vec{S} \cdot \vec{S} = 1$. In this formulation, the local current densities $j_{x,y}^{ab}$ in the field theory:[5]

$$j_x^{ab} = \left[ S^a(x,y)\partial_x S^b(x,y) - S^b(x,y)\partial_x S^a(x,y) \right] , \tag{99a}$$

$$j_y^{ab} = \left[ S^a(x,y)\partial_y S^b(x,y) - S^b(x,y)\partial_y S^a(x,y) \right] , \tag{99b}$$

can be studied at long distance by considering the lattice quantities

$$j_x^{ab} \approx \frac{1}{\epsilon} \left[ S_{(i,j)}^a S_{(i+1,j)}^b - S_{(i,j)}^b S_{(i+1,j)}^a \right] , \tag{100a}$$

$$j_y^{ab} \approx \frac{1}{\epsilon} \left[ S_{(i,j)}^a S_{(i,j+1)}^b - S_{(i,j)}^b S_{(i,j+1)}^a \right] , \tag{100b}$$

where $\epsilon$ is the lattice spacing - set equal to unity in what follows, and $(i,j)$ are the lattice coordinates. Notice the normalization is the same as in (94,95).

---

[5]The chiral components are $j = j_x - ij_y$ and $\bar{j} = -(j_x + ij_y)$, since in our conventions $\bar{\partial} j = \partial \bar{j}$.

Consider now the correlation function

$$\left\langle j_y^{ab} j_{y'}^{ba} \right\rangle = \left\langle \left( S_{(i,j)}^a S_{(i,j+1)}^b - S_{(i,j)}^b S_{(i,j+1)}^a \right) \left( S_{(i',j')}^b S_{(i',j'+1)}^a - S_{(i',j')}^a S_{(i',j'+1)}^b \right) \right\rangle . \tag{101}$$

Inserting this into the high-temperature expansion of the model, we now get a modified partition function where, in addition to loops we have *either* a line connecting (a contraction) $(ij)$ and $(i', j'+1)$ and carrying the label $a$ (resp. $b$), and one connecting $(i, j+1)$ and $(i', j')$ and carrying the label $b$ (resp. $a$), *or* a line connecting $(ij)$ and $(i', j')$ and carrying the label $a$ (resp. $b$), and one connecting $(i, j+1)$ and $(i', j'+1)$ and carrying the label $b$ (resp. $a$). The latter situation in either case comes with a minus sign. For instance,

$$S_{(i,j)}^a S_{(i,j+1)}^b S_{(i',j')}^b S_{(i',j'+1)}^a : \tag{102a}$$

$$S_{(i,j)}^a S_{(i,j+1)}^b S_{(i',j')}^a S_{(i',j'+1)}^b : \tag{102b}$$

It follows from this discussion that the current-current two-point function in the $O(n)$ (101) can be reformulated as the difference between two well-defined geometrical partition functions with lines connecting the insertions as in (102). Of course, this can be re-interpreted in terms of orientation correlations, as we shall now see.

## 6.3  $U(1)$ and loop correlators

We now observe, following [1,2], that we could decide to orient the loops and give each oriented loop a fugacity $\frac{n}{2}$ without changing the partition function. This would correspond to considering instead of a real $O(n)$ model a complex $O(\frac{n}{2})$ one, with a modified lattice interaction

$$2\vec{S}_i \cdot \vec{S}_j \to \vec{s}_i^{\,*} \cdot \vec{s}_j + \text{herm. conj.}, \tag{103}$$

with $\vec{s}$ denoting complex vectors with $\frac{n}{2}$ components and unit length. This model enjoys now a global $U(1)$ symmetry under $\vec{s} \to e^{i\phi}\vec{s}$, and the associated currents read

$$j_x \propto \lim_{\epsilon \to 0} \frac{1}{\epsilon} \sum_\alpha \left[ (s_{(i,j)}^\alpha)^* s_{(i+1,j)}^\alpha - s_{(i,j)}^\alpha (s_{(i+1,j)}^\alpha)^* \right] = \sum_\alpha (s^\alpha)^* \partial_x s^\alpha - s^\alpha \partial_x (s^\alpha)^* , \tag{104a}$$

$$j_y \propto \lim_{\epsilon \to 0} \frac{1}{\epsilon} \sum_\alpha \left[ (s_{(i,j)}^\alpha)^* s_{(i,j+1)}^\alpha - s_{(i,j)}^\alpha (s_{(i,j+1)}^\alpha)^* \right] = \sum_\alpha (s^\alpha)^* \partial_y s^\alpha - s^\alpha \partial_y (s^\alpha)^* , \tag{104b}$$

where the sums for $\alpha$ run from 1 to $\frac{n}{2}$. The geometrical interpretation of the correlation function (note that $\alpha \neq \beta$ components do not couple)

$$\left\langle \sum_\alpha \left( (s_{(i,j)}^\alpha)^* s_{(i,j+1)}^\alpha - s_{(i,j)}^\alpha (s_{(i,j+1)}^\alpha)^* \right) \left( (s_{(i',j')}^\alpha)^* s_{(i',j'+1)}^\alpha - s_{(i',j')}^\alpha (s_{(i',j'+1)}^\alpha)^* \right) \right\rangle , \tag{105}$$

is now that we have *either* a line connecting $(i, j)$ and $(i', j'+1)$ with a certain orientation, and one connecting $(i, j+1)$ and $(i', j')$ with the opposite orientation, *or* a line connecting $(i, j)$ and $(i', j')$ with a certain orientation, and one connecting $(i, j+1)$ and $(i', j'+1)$ with

the opposite orientation, this latter case coming with a minus sign. This is illustrated below with the black lines, which now carry an orientation from $s$ to $s^*$:

$$(s_{(i,j)}^{\alpha})^* s_{(i,j+1)}^{\alpha} (s_{(i',j')}^{\alpha})^* s_{(i',j'+1)}^{\alpha} : \qquad \qquad , \qquad (106a)$$

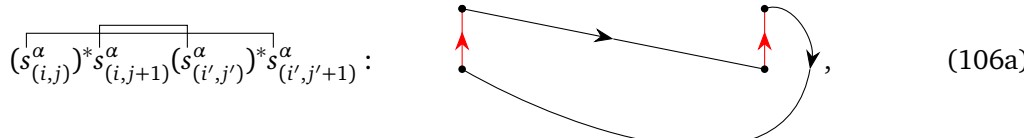

$$(s_{(i,j)}^{\alpha})^* s_{(i,j+1)}^{\alpha} s_{(i',j')}^{\alpha} (s_{(i',j'+1)}^{\alpha})^* : \qquad \qquad . \qquad (106b)$$

Clearly, this is identical to the object we defined in (101)—although in the second case we get a multiplicity $\frac{n}{2}$ corresponding to the sum over all possible colors $\alpha$ (we do not sum over colors in (101). It follows that

$$\langle j_\mu^{ab} j_\mu^{ba} \rangle = \frac{2}{n} \langle j_\mu j_\mu \rangle. \qquad (107)$$

The pictures (106) can be completed by adding red lines inside each insertion of the lattice current operator, now oriented from $s^*$ to $s$. As a result, each of the contributions to the current-current correlation function is represented as an oriented loop. One may then notice that $\langle j_\mu j_\mu \rangle$ can be interpreted as the the probability that the edges $(i,j)(i,j+1)$ and $(i',j')(i',j'+1)$—now considered as having a fixed orientation along the $y$-direction—belong to the same loop and are traversed in the same direction, minus the probability that they belong to the same loop but are traversed in opposite directions. In other words, $\langle j_\mu j_\mu \rangle$ is a two-point correlation function of *orientations* of loops. As such, this quantity has been of interest in various contexts. In particular, in [2, 9], the authors were able to determine, using Coulomb gas techniques, the long-distance behavior of (107). Setting

$$\langle j(z,\bar{z}) j(w,\bar{w}) \rangle = \frac{\kappa}{(z-w)^2}, \qquad (108)$$

and the same for $\bar{j}$, it was found in [9] that[6]

$$\kappa = \frac{(\beta^2 - 1)}{8\pi\beta^2} \cot(\pi\beta^2). \qquad (109)$$

Remarkably, this is in agreement with the exact bootstrap results, at the price of some change of normalization, since we have

$$\frac{k}{(2\pi)^2} = \frac{2}{n}\kappa, \qquad (110)$$

where $k$ comes from (75) and $\kappa$ from (109). The factor $\frac{2}{n}$ on the right-hand side arises from the one in (107). The factors of $2\pi$ are a matter of convention: the normalization of the currents that we have chosen in the CFT part of this paper is such that integrated charges satisfy the $O(n)$ Lie-algebra relations with structure constants obeying $|f_C^{AB}| = 1$. Referring, e.g., to equation (25) we see that, on a cylinder (of circumference $2\pi$ by convention, as usual), we have

$$J_0^A = \frac{1}{2i\pi} \int_{-\pi}^{\pi} J^A(\sigma)\, d\sigma. \qquad (111)$$

On the other hand, in (94) and (101), we have used the statistical physics convention to define current densities such that their integral (without factors of $2\pi$) gives the conserved charges. In other words, we must identify $\frac{J^A}{2\pi}$ with $j^{ab}$, which introduces a factor $\frac{1}{(2\pi)^2}$ in (110) when comparing the two-point functions.

---

[6]Actually, the constant called $\kappa$ in [9] differs from this by a factor 16 due to a different normalization: see the appendix for more detail.

# 7 Conclusion

Properties of currents may have further interesting applications in the case of models with $U(n)$ symmetry, relevant in particular to the study of hulls in the $Q$-state Potts model, and potentially the plateau transition in the class-C spin quantum Hall effect [37]—we hope to report on this in the near future.

From a more fundamental point of view, currents should play more than an anecdotic (albeit of high physical interest) role in the analysis of the $O(n)$ CFT. Indeed, much of the earlier progress on this problem took place using the Coulomb gas formalism where, in particular, the concept of charge ("electric" or "magnetic") of physical observables, combined with the presence of a "charge at infinity", played a crucial role. While the bootstrap makes these considerations seem irrelevant, it is intriguing to wonder how much of the Coulomb gas approach can be salvaged—after all, this approach remains crucial to determine the spectrum of the theory.

We will content ourselves with a simple observation here. Focussing only on fields with dimension $(2, 0)$, we have the OPE

$$J(z)J(w) = \frac{\kappa}{(z-w)^2}\left[V^D_{\langle 1,1\rangle} + (z-w)^2\left(\frac{2}{c}L_{-2}V^D_{\langle 1,1\rangle} + \frac{C_{JJ\partial J}}{\kappa}\partial J\right) + \dots\right]$$
$$= \frac{\kappa}{(z-w)^2}\left[V^D_{\langle 1,1\rangle} + (z-w)^2\left(\frac{2}{c}T + \frac{1}{2\kappa}\partial J\right) + \dots\right], \tag{112}$$

where we used $L_{-2}V^D_{\langle 1,1\rangle} = T$ and normalized the current as usual, so that $C_{JJJ} = 1$. The coefficients of $T$ and $\partial J$ are completely fixed by general Ward identities. Of course we know that the OPE should also contain infinitely many logarithmic terms in $\ln(z-w)$ and $\ln(\bar{z}-\bar{w})$, but this will not matter for us here.

We now define normal-ordering as usual, by subtracting all terms in the OPE that are singular as $z \to w$. It follows that

$$:JJ:(z) = \frac{2\kappa}{c}T + \frac{1}{2}\partial J. \tag{113}$$

We can now project this equation onto the $O(n)$-singlet sector to obtain

$$T(z) = \frac{c}{2\kappa}:JJ:(z)\Big|_{[\,]}. \tag{114}$$

The notation is of course compact. By giving indices to the currents, the right-hand side can be interpreted as the usual $O(n)$ quadratic Casimir contraction: we see therefore that the stress-energy tensor in the theory has exactly the form one would expect for a WZW theory. This happens simply because $T$ is forced to appear in the $JJ$ OPE due to conformal Ward identities, and there is no other field with weights $(2, 0)$ in the spectrum with the right symmetry.

It is tempting now to define charges via the OPEs of fields with the currents, and to interpret the conformal weights obtained from the Coulomb gas formalism in the light of (114). Of course, the currents being non-chiral, considerable care has to be exercised in extending the well-known analysis that would hold in a WZW theory (or, more simply, a $U(1)$ free-boson theory as in [2]) to the $O(n)$ CFT: for instance, conformal weights are not simply squares of the charges. How this works out—and how the charge at infinity appears—will be discussed elsewhere.

## Acknowledgments

We thank S. Ribault for related collaborations, many illuminating discussions, and a careful reading of the manuscript. We thank J. Cardy for useful discussion, and his kind interest in our

work. We are also grateful to the two referees of SciPost for comments that helped us improve the presentation of the paper.

**Funding information** This work was supported by the French Agence Nationale de la Recherche (ANR) under grant ANR-21-CE40-0003 (project CONFICA).

# A Diagrams and the four-point correlation function of currents

The goal of this appendix is to prove equation (72). This involves diagrammatic interpretations of various $O(n)$ algebraic objects, strongly inspired by the study of "bird-tracks" in [38, 39].

## A.1 Projectors

We use lower-case Latin letters to denote the $n$ states in the fundamental $O(n)$ representation. We also use a ket notation, so $[1] = \mathrm{Span}\{|a\rangle, a = 1, \ldots, n\}$. The tensor product of two fundamentals decomposes as $[1] \times [1] = [\,] + [11] + [2]$, and we introduce the corresponding projectors:

$$P_{[2]}|ab\rangle = \frac{|ab\rangle + |ba\rangle}{2} - \frac{\delta_{ab}}{n}\sum_c |cc\rangle, \tag{A.1a}$$

$$P_{[11]}|ab\rangle = \frac{|ab\rangle - |ba\rangle}{2}, \tag{A.1b}$$

$$P_{[\,]}|ab\rangle = \frac{\delta_{ab}}{n}\sum_c |cc\rangle. \tag{A.1c}$$

It will be useful in what follows to introduce the notation

$$|(ab)\rangle \equiv \frac{|ab\rangle - |ba\rangle}{2}. \tag{A.2}$$

We now consider the tensor product of two adjoint representations. Reorganizing (9) we have

$$[11] \times [11] = [1111] + ([22] + [2] + [\,]) + ([211] + [11]) \tag{A.3}$$

(the parentheses will be explained shortly), and our goal is to write the projectors onto all the representations on the right-hand side.

It is useful to start by considering the product in $SU(n)$:

$$[11]_{su} \times [11]_{su} = [1111]_{su} + [22]_{su} + [211]_{su}, \tag{A.4}$$

and observe that the representation $[11]$ has the same basis for both algebras, with $\frac{n(n-1)}{2}$ states $|(ab)\rangle$—we label identically the basis states in the fundamental representation for the two cases. The parentheses in (A.3) then simply indicate the branching of $SU(n)$ into $O(n)$ representations. In the tensor product (A.4), the first two representations are in the symmetric (S) sector under the exchange of the two $[11]$, while the last one is in the antisymmetric (A) sector. These sectors are easily obtained via

$$P_{S,A}|abcd\rangle = P_{S,A}|(ab)(cd)\rangle = \frac{1}{2}\big(|(ab)(cd)\rangle \pm |(cd)(ab)\rangle\big). \tag{A.5}$$

Projection on $[211]$ in $SU(n)$ is therefore immediate:

$$P_{[211]_{su}}|(ab)(cd)\rangle = \frac{1}{2}\big(|(ab)(cd)\rangle - |(cd)(ab)\rangle\big). \tag{A.6}$$

All we have to do now is to subtract the trace to obtain the corresponding projector in $O(n)$:

$$P_{[211]}|(ab)(cd)\rangle = \frac{1}{2}\big(|(ab)(cd)\rangle - |(cd)(ab)\rangle\big)$$
$$-\frac{1}{2(n-2)}\Bigg[\delta_{bc}\sum_e\big(|(ae)(ed)\rangle - |(ed)(ae)\rangle\big) - \delta_{bd}\sum_e\big(|(ae)(ec)\rangle - |(ec)(ae)\rangle\big)$$
$$-\delta_{ac}\sum_e\big(|(be)(ed)\rangle - |(ed)(be)\rangle\big) + \delta_{ad}\sum_e\big(|(be)(ec)\rangle - |(ec)(be)\rangle\big)\Bigg].$$
$$(A.7)$$

The projector onto the remaining adjoint,

$$P_{[11]} = \frac{1}{2(n-2)}\Bigg[\delta_{bc}\sum_e\big(|(ae)(ed)\rangle - |(ed)(ae)\rangle\big) - \delta_{bd}\sum_e\big(|(ae)(ec)\rangle - |(ec)(ae)\rangle\big)$$
$$-\delta_{ac}\sum_e\big(|(be)(ed)\rangle - |(ed)(be)\rangle\big) + \delta_{ad}\sum_e\big(|(be)(ec)\rangle - |(ec)(be)\rangle\big)\Bigg],$$
$$(A.8)$$

follows from $P_{[211]} + P_{[11]} = P_{[211]_{su}}$.

For the fully anti-symmetric sector, we can of course immediately write the projector onto the $[1111]$ representation:

$$P_{[1111]}|(ab)(cd)\rangle = P_{[1111]_{su}}|(ab)(cd)\rangle$$
$$= \frac{1}{6}\big(|(ab)(cd)\rangle + |(cd)(ab)\rangle - |(ac)(bd)\rangle$$
$$- |(bd)(ac)\rangle + |(bc)(ad)\rangle + |(ad)(bc)\rangle\big).\qquad(A.9)$$

We also observe that

$$P_{[1111]}|(ab)(cd)\rangle = P_{[1111]}|abcd\rangle.\qquad(A.10)$$

The projector for $SU(n)$ then follows (since the first two terms in (A.4) make up the full symmetric sector):

$$P_{[22]_{su}}|(ab)(cd)\rangle = P_S|(ab)(cd)\rangle - P_{[1111]}|(ab)(cd)\rangle.\qquad(A.11)$$

For completeness, we write the corresponding expression:

$$P_{[22]_{su}}|(ab)(cd)\rangle = \frac{1}{3}\big[|(ab)(cd)\rangle + |(cd)(ab)\rangle\big]$$
$$+ \frac{1}{6}\big[|(ac)(bd)\rangle + |(bd)(ac)\rangle - |(bc)(ad)\rangle - |(ad)(bc)\rangle\big].\qquad(A.12)$$

Meanwhile we can consider the double trace, which is the projector onto the $O(n)$ identity:

$$P_{[\,]}|(ab)(cd)\rangle = \frac{2}{n(n-1)}\big(\delta_{bc}\delta_{ad} - \delta_{ac}\delta_{bd}\big)\sum_{e,f}|(ef)(fe)\rangle.\qquad(A.13)$$

Exchanging $e \leftrightarrow f$ shows that this is in the symmetric sector indeed, while the fact that we act on $|(ab)(cd)\rangle$ fixes the sign in the delta functions. Next, we can build by inspection the

projector onto the representation $[2]$:

$$P_{[2]}|(ab)(cd)\rangle = \frac{1}{2(n-2)}\left[\delta_{bc}\left(\sum_e\left(|(ae)(ed)\rangle + |(ed)(ae)\rangle\right)\right)\right.$$

$$-\delta_{ac}\left(\sum_e\left(|(be)(ed)\rangle + |(ed)(be)\rangle\right)\right) - \delta_{bd}\left(\sum_e\left(|(ae)(ec)\rangle + |(ec)(ae)\rangle\right)\right)$$

$$+\left.\delta_{ad}\left(\sum_e\left(|(be)(ec)\rangle + |(ec)(be)\rangle\right)\right) - \frac{4}{n}(\delta_{bc}\delta_{ad} - \delta_{ac}\delta_{bd})\sum_{e,f}|(ef)(fe)\rangle\right], \quad \text{(A.14)}$$

where the last term is there to ensure tracelessness. This leaves finally $P_{[22]} = P_{[22]_{su}} - P_{[2]} - P_{[\,]}$:

$$P_{[22]}|(ab)(cd)\rangle = \frac{1}{3}\left[|(ab)(cd)\rangle + |(cd)(ab)\rangle\right]$$

$$+ \frac{1}{6}\left[|(ac)(bd)\rangle + |(bd)(ac)\rangle - |(bc)(ad)\rangle - |(ad)(bc)\rangle\right]$$

$$-\frac{1}{2(n-2)}\left[\delta_{bc}\left(\sum_e|(ae)(ed)\rangle + |(ed)(ae)\rangle\right) - \delta_{ac}\left(\sum_e|(be)(ed)\rangle + |(ed)(be)\rangle\right)\right.$$

$$\left.-\delta_{bd}\left(\sum_e|(ae)(ec)\rangle + |(ec)(ae)\rangle\right) + \delta_{ad}\left(\sum_e|(be)(ec)\rangle + |(ec)(be)\rangle\right)\right]$$

$$+ \frac{1}{(n-1)(n-2)}(\delta_{bc}\delta_{ad} - \delta_{ac}\delta_{bd})\sum_{e,f}|(ef)(fe)\rangle. \quad \text{(A.15)}$$

In conclusion, we have built explicitly all the projectors for the product $[11]\times[11]$ in (A.3). We note that a bird-track version of this decomposition can be found in [39].

## A.2 Four point-functions

We now derive the relation between the $O(n)$ tensors and the diagrams for four-point functions of $O(n)$ vectors and $O(n)$ adjoints.

### A.2.1 $O(n)$ vectors

Before moving to the case of primary interest—that of adjoints—we first study a simpler example to fix ideas. We consider the four-point function $\langle V_{(\frac{1}{2},0)}V_{(\frac{1}{2},0)}V_{(\frac{1}{2},0)}V_{(\frac{1}{2},0)}\rangle$. Recall that $V_{(\frac{1}{2},0)}$ transforms as an $O(n)$ vector. We therefore label this four-point function with $O(n)$ tensor indices as follows:

$$\left\langle \left(V_{(\frac{1}{2},0)}\right)_a\left(V_{(\frac{1}{2},0)}\right)_b V^c_{(\frac{1}{2},0)}V^d_{(\frac{1}{2},0)}\right\rangle = G^{cd}_{ab} = P_{[2]}F^{[2]} + P_{[11]}F^{[11]} + P_{[\,]}F^{[\,]}, \quad \text{(A.16)}$$

where on the right-hand side we have made a decomposition, according to the $O(n)$ tensor product $[1]\times[1] = [2] + [11] + [\,]$ (see Section 2.1 of [17]), in terms of projectors $P_\lambda$ on irreducibles $\lambda$ and the corresponding coefficients $F^\lambda$. Using formulas (A.1) we have that

$$\delta_{ab}\delta^{cd} = n(P_{[\,]})^{cd}_{ab}, \quad \text{(A.17a)}$$

$$\delta^d_a\delta^c_b = P = (P_{[2]} - P_{[11]} + P_{[\,]})^{cd}_{ab}, \quad \text{(A.17b)}$$

$$\delta^c_a\delta^d_b = I = (P_{[2]} + P_{[11]} + P_{[\,]})^{cd}_{ab}. \quad \text{(A.17c)}$$

This leads to

$$(P_{[2]})_{ab}^{cd} = \frac{1}{2}\left(\delta_a^c\delta_b^d + \delta_a^d\delta_b^c\right) - \frac{1}{n}\delta_{ab}\delta^{cd},$$

$$(P_{[11]})_{ab}^{cd} = \frac{1}{2}\left(\delta_a^c\delta_b^d - \delta_a^d\delta_b^c\right). \tag{A.18}$$

So we can rewrite the correlator (A.16) as

$$G_{ab}^{cd} = \delta_{ab}\delta^{cd}\frac{F^{[\,]} - F^{[2]}}{n} + \delta_a^d\delta_b^c\frac{F^{[2]} - F^{[11]}}{2} + \delta_a^c\delta_b^d\frac{F^{[2]} + F^{[11]}}{2}. \tag{A.19}$$

By convention, we associate with the amplitude of every product of Kronecker deltas a diagram, $G_i$ with $i = 1, 2, 3$, defined as follows:

$$\delta_{ab}\delta^{cd} \mapsto G_1, \qquad\qquad \delta_a^d\delta_b^c \mapsto G_2, \qquad\qquad \delta_a^c\delta_b^d \mapsto G_3. \tag{A.20}$$

In our convention, the $O(n)$ labels of the points $1, 2, 3, 4$ are $a, b, c, d$ in that order. Therefore we can write

$$G_{ab}^{cd} = G_1\delta_{ab}\delta^{cd} + G_2\delta_a^d\delta_b^c + G_3\delta_a^c\delta_b^d. \tag{A.21}$$

Comparing with (A.17), we finally find the formula

$$G = 2P_{[2]}\left(\frac{G_2 + G_3}{2}\right) + 2P_{[11]}\left(\frac{-G_2 + G_3}{2}\right) + nP_{[\,]}\left(G_1 + \frac{G_2 + G_3}{n}\right), \tag{A.22}$$

where the appearance of the projectors on the right-hand side must be interpreted by writing the components of $G$ in terms of matrix elements of the $P$'s. Alternatively, it is sometimes useful to think of $G$ as an operator "propagating in the s-channel"—here acting on the states $|ab\rangle$, with $G|ab\rangle = \sum_{c,d} G_{ab}^{cd}|cd\rangle$—, though we will not use a separate notation for this.

Formula (A.22) can in fact be found in a different, more useful way. We suppose we know (A.21) and wish, e.g., to find $F_{[\,]}$ in (A.16). We isolate its contribution by computing the projection $GP_{[\,]}$:

$$\begin{aligned} GP_{[\,]}|ab\rangle &= \delta_{ab}\frac{1}{n}G\sum_c|cc\rangle = \delta_{ab}\frac{1}{n}\sum_{c,d}G_{cc}^{dd}|dd\rangle = \delta_{ab}\frac{1}{n}\sum_c\left(\sum_{d\neq c}G_{cc}^{dd}|dd\rangle + G_{cc}^{cc}|cc\rangle\right) \\ &= \delta_{ab}\frac{1}{n}\sum_d\left[(n-1)G_1|dd\rangle + (G_1 + G_2 + G_3)|dd\rangle\right] \\ &= (nG_1 + G_2 + G_3)P_{[\,]}|ab\rangle. \end{aligned} \tag{A.23}$$

So we find $F_{[\,]} = nG_1 + G_2 + G_3$, which is indeed in agreement with the coefficient of $P_{[\,]}$ in (A.22). Note this result can in fact be obtained more quickly. Since $GP_{[\,]} \propto P_{[\,]}$ all we have to do to determine the proportionality coefficient is to consider $GP_{[\,]}|aa\rangle$ and extract the component proportional to $|aa\rangle$. $P_{[\,]}|aa\rangle$ is given by $\frac{1}{n}\sum_c|cc\rangle$. We consider all the choices $aa, cc$ and draw all the possible diagrams compatible with these colors. If $c \neq a$ we have one diagram $G_1$, providing overall $\frac{n-1}{n}G_1$. If $c = a$ we can still have $G_1$, so in fact $G_1$ comes with a factor $\frac{n-1}{n} + \frac{1}{n} = 1$. But if $c = a$ we can also have $G_2$ and $G_3$, each coming with a factor $\frac{1}{n}$. Hence the amplitude is proportional to $G_1 + \frac{G_2 + G_3}{n}$.

### A.2.2 $O(n)$ adjoints

We now consider a four-point function involving four fields, which transforms only in the adjoint of $O(n)$—the so-called currents. Such an object requires a priori the introduction of indices:

$$G_{(ab)(cd)}^{(ef)(gh)} = \langle J_{(ab)} J_{(cd)} J^{(ef)} J^{(gh)} \rangle . \tag{A.24}$$

It is convenient to factor out the dependency on these indices and write, in analogy with the right-hand side of (A.16),

$$G = \langle JJJJ \rangle = P_{[1111]} F_{(s)}^{[1111]} + P_{[22]} F_{(s)}^{[22]} + P_{[2]} F_{(s)}^{[2]} + P_{[\,]} F_{(s)}^{[\,]} + P_{[211]} F_{(s)}^{[211]} + P_{[11]} F_{(s)}^{[11]} , \tag{A.25}$$

where now we have used the $O(n)$ tensor product (9). Here, $F_{(s)}^{\lambda}$ are solutions to the crossing-symmetry equations in the $s$-channel, $P_{\lambda}$ are $O(n)$ projectors, and the object on the left-hand side is now thought of—see the remark after (A.22)—as an operator taking the indices of the first two insertions to the ones of the last two ones, so

$$G|(ab)(cd)\rangle = \sum_{(ef)(gh)} G_{(ab)(cd)}^{(ef)(gh)} |(ef)(gh)\rangle . \tag{A.26}$$

We now go back to the four-point correlator of the currents and study first the component onto the fully antisymmetric representation, $\langle JJJJ \rangle_{[1111]}$. The question is how to interpret this component in terms of diagrams.

To do this, just like in the simpler case of $\langle V_{(\frac{1}{2},0)} V_{(\frac{1}{2},0)} V_{(\frac{1}{2},0)} V_{(\frac{1}{2},0)} \rangle$, we consider $GP_{[1111]}|(ab)(cd)\rangle$ and extract the component proportional to $|(ab)(cd)\rangle$. To proceed, we define *signed diagrams* as follows. We split the points representing the operator insertions (in $1, 2, 3, 4$) in as many points as there are colors, and assign to these points colors in the same order as in the associated ket. We then imagine drawing, starting from every such point, a (colored) dotted line going to $-\infty$ for the insertions $1, 2$ and $+\infty$ for the insertions $3, 4$. Finally, we connect identical colors (since $a, b, c, d$ are all different by construction no ambiguity arises), and define an index $p$ as the number of intersections of full lines with dotted lines of a different color. The value of the diagram is given by $(-1)^p$ times the sum of the usual $O(n)$ Boltzmann weights over loop configurations represented in the diagram—see the illustrations in (A.28) below.

Now $P_{[1111]}|(ab)(cd)\rangle$ is given in (A.9). Imposing that the points $3, 4$ are in the state $|(ab)(cd)\rangle$ gives therefore a total of $6 \times 2^4$ terms. The multiplicity $6$ comes from the antisymmetrizations on the right-hand side of (A.9) and the multiplicity $2^4$ arises since, for each point $1, 2, 3, 4$ we can permute the two corresponding colors. The amplitude we are looking for, $F^{[1111]}$, is proportional to the sum of all diagrams corresponding to the terms in (A.9). Up to the detail of connectivities at the split extremities, we get the diagrams $K_2, K_3$ and $L_3$ out of the possible diagrams in (36). They come with relative multiplicities $1, 1, 4$ (corresponding to the 6 terms in (A.9)). It follows that

$$F_{(s)}^{[1111]} \propto F_{(s)}^{K_2} + F_{(s)}^{K_3} + 4 F_{(s)}^{L_3} . \tag{A.27}$$

Note that in fact, when writing this, we should really specify that each type of diagram should be "decorated" by $2^4$ different assignments of colors at its extremities, and the corresponding "point-split" diagram evaluated with the rules discussed earlier. In particular, $G_{(ab)(cd)}^{(ab)(cd)}$ and $G_{(ab)(cd)}^{(cd)(ab)}$ correspond to

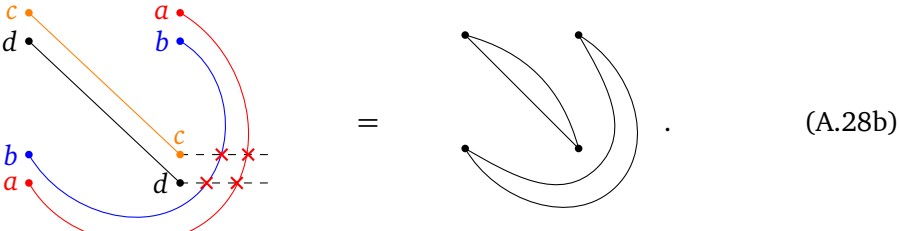

$$\text{(A.28a)}$$

$$\text{(A.28b)}$$

$G_{(ab)(cd)}^{(ac)(bd)}$ , $G_{(ab)(cd)}^{(bc)(ad)}$ , $G_{(ab)(cd)}^{(bd)(ac)}$ , $G_{(ab)(cd)}^{(ad)(bc)}$ correspond to the signed diagrams:

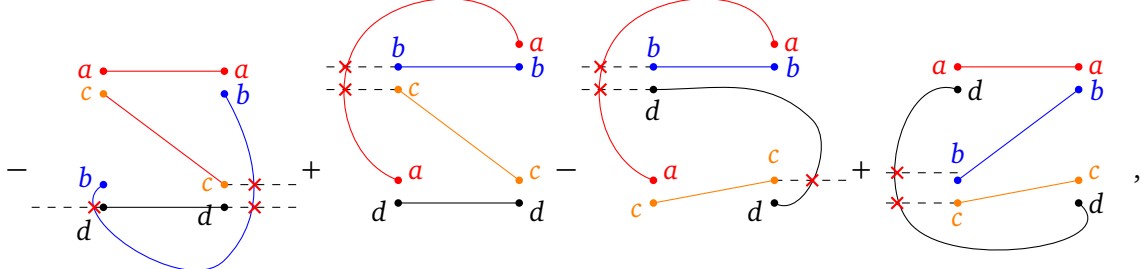

so, after combining with the signs in (A.9) we obtain

$$4 \times \qquad . \qquad\qquad \text{(A.29)}$$

Adding this to the two terms on the right hand side of eq. (A.28) reproduces (A.27) indeed.

# B  A numerical determination of $k$ using transfer matrices

The purpose of this Appendix is to construct the two-point function of the current operator directly in the lattice model and to study numerically some of its properties. In particular we shall compute a finite-size approximation to the level parameter $k$ and extrapolate it to the thermodynamic limit. It will become apparent that the lattice constructions can be generalized in various ways, e.g. to the case of higher-point correlation functions.

## Lattice model

For definiteness we consider Nienhuis' O($n$) model on the hexagonal lattice:

$$ \text{(B.1)} $$

The configurations are sets of self-avoiding and mutually avoiding closed loops on this lattice, obtained by occupying a subset of the edges by monomers—shown below in blue color—so each vertex is incident on zero or two monomers. The partition function is then

$$ Z(K,n) = \sum_{\text{loops}} K^{\#\text{monomers}} n^{\#\text{loops}} . \tag{B.2} $$

The monomer fugacity at the critical point is known to be [40]

$$ K_c = \left( 2 \pm \sqrt{2-n} \right)^{-1/2} , \tag{B.3} $$

where $-2 \leq n \leq 2$, and we take the plus (minus) sign for the dilute (dense) phase.

## Transfer matrix

To construct a time-evolution operator (transfer matrix) we pick a distinguished "time" direction, by convention taken to be upwards. The orthogonal, horizontal direction then defines the "space" direction. Notice that we have oriented the lattice (B.1) so that one third of the edges are parallel to the space direction. The mid-points of the remaining, slanted edges are called sites, and a horizontal (i.e., space-like) line intersecting $2L$ sites is called a time slice. We label the sites within a time slice $i = 1, 2, \ldots, 2L$, from left to right. The figure (B.1) thus shows a lattice of width $L = 4$.

The goal of the transfer matrix construction is to build up a semi-infinite cylinder of circumference $L$, obtained by imposing periodic boundary conditions in the space direction and taking the limit of infinite height along the time direction. To this end we first define the so-called $\check{R}$ matrix that builds up a small piece of the lattice

$$ \check{R}_k = \underset{k \quad k+1}{\bigtimes} , \tag{B.4} $$

by acting on the sites labelled $k$ and $k+1$. The corresponding sum over loop configurations can be depicted as

$$ \check{R}_k = \bigtimes + K\bigtimes + K\bigtimes + K^2\underset{\text{cup}}{\smile} + K^2\diagup + K^2\diagdown + K^2\bigtimes + K^2\underset{\text{cap}}{\frown} , \tag{B.5} $$

where we have shown in front of each diagram the local part of the Boltzmann weight, accounting for the number of monomers. Notice that $\check{R}_k$ constructs one horizontal edge and four slanted half-edges. Two of the diagrams have been given convenient nicknames, "cup" and "cap", which will be used below.

The row-to-row transfer matrix for a system of width $L$ builds a whole layer of the lattice, meaning that it transfers from one time slice to the next. It can be written

$$T = \left(\prod_{j=1}^{L} \check{R}_{2j-1}\right) \times \left(\prod_{j=1}^{L} \check{R}_{2j}\right), \tag{B.6}$$

where we have identified the site labels modulo $2L$ in order to have periodic boundary conditions.

Suppose first that the goal was to construct the partition function (B.2). The space of states on which $T$ acts would then be the set of dilute defect-free link patterns over the set of sites $\{1, 2, \ldots, 2L\}$. Such a link pattern is a collection of $p$ arcs with $0 \leq p \leq L$, such that each arc connects two distinct sites, each site connects to at most one arc, and arcs do not cross. The link patterns can be depicted by drawing the $p$ non-intersecting arcs in the half-space below the time slice. For example, here is a dilute link pattern with $p = 4$ arcs in the case $L = 6$:

$$\tag{B.7}$$

The link patterns contain precisely the information necessary to compute the non-local part of the Boltzmann weight, accounting for the number of loops. Namely, in (B.5), four of the diagrams act trivially on the link patterns, another two just make one end of an arc jump one site to the left or right, and the cup replaces two adjacent empty sites by an arc. The most non-trivial action is provided by the cap, which can either concatenate two distinct arcs, or, if both sites belong to the same arc, register the completion of a loop and provide the corresponding weight $n$.

The transfer matrix is also useful for building correlation functions. The simplest correlation function is the (unnormalized) probability that $\ell$ open, non-intersecting paths extend from the bottom to the top time slice of a finite-height cylinder. It can be computed by letting $T$ act on link patterns with $\ell$ defects—often called through-lines—which can move around in the same way as the loops, but which are not allowed to undergo pairwise annihilation under the action of the cap operator.

**Spectrum**

The spectrum of $T$ within the basis of all link patterns can be decomposed with respect to the number of through-lines $\ell$ (with $\ell = 0, 1, \ldots, 2L$), the momentum of through-lines $k$ (with $k = 0, 1, \ldots, \ell - 1$) and the lattice momentum $m$ (with $m = 0, 1, \ldots, L - 1$). We denote the corresponding sets of eigenvalues by $V_{\ell,k,m}$, as in Appendix A of [41]. To obtain this decomposition it is important to notice that the lattice (B.1) is invariant under a horizontal shift by *two* sites. Accordingly $T$ commutes with the two-step shift operator $u^2$. The quantization of $k$ results from the observation, that the link pattern is unchanged if all $\ell$ through-lines are brought around the periodic boundary condition and back to their initial positions. The lattice momentum $m$ follows from the simultaneous diagonalization of $T$ and $u^2$, and it is quantized by the periodic boundary conditions.

A first technical step is to write an efficient code that extracts the transfer matrix $T_{\ell,k,m}$ in each sector. This follows Appendix A.4 of [41], the main change being that the basis states are now link patterns of the dilute O($n$) model, rather than the completely packed link patterns used in [41] to deal with the Potts model. The momentum sectors are constructed by selecting a representative state for each orbit of the link patterns under the cyclic group generated by $u^2$ and packing (unpacking) the link patterns to (from) this smaller space just after (before)

each action by $T$ by means of an operator $S_{\text{out}}$ ($S_{\text{in}}$). This procedure is described in details in Appendix A.4 of [41].

We have checked that the spectrum of $T$ is indeed the union of the resulting spectra $V_{\ell,k,m}$. A novel feature, not encountered in the study of the Potts model, is that some of the eigenvalues are not real, but appear in complex conjugate pairs. This is however still compatible with the partition function $Z(K,n)$ and the defect-path correlation functions being real.

### Observable

Our goal is now to make the discussion of Section 6.3 amenable to the transfer-matrix setup. We wish to compute a correlation function, in which two specific edges are required to be traversed by the same loop, giving different signs to the two different relative directions of traversal (106).

To this end we first mark two space-like (i.e., horizontal) edges at the same space coordinate and separated by $t$ rows in the time direction, corresponding to the action of $T^t$. We orient both these edges from left to right.

We next define a variant transfer matrix $\widetilde{T}$ that builds only those configurations in which the two marked edges are forced to be occupied and to be traversed by the *same* loop. We call this the marked loop; it still has a weight $n$. With this constraint we wish to build four distinct partition functions $\widetilde{Z}_\alpha(K,n)$. The cases $\alpha = 1,2,3,4$ imply the following characterisation of the marked loop:

- For $\alpha = 1$ it is *contractible*, and following this loop we pass one of the marked edges in its chosen direction and the other one in the *opposite* direction.

- For $\alpha = 2$ it is *contractible* and the passage directions are *identical*.

- For $\alpha = 3$ it is *non-contractible* and the passage directions are *opposite*.

- For $\alpha = 4$ it is *non-contractible* and the passage directions are *identical*.

Notice that the destinction between these four cases does not depend on the direction in which we follow the marked loop. The following figure illustrates the four cases:

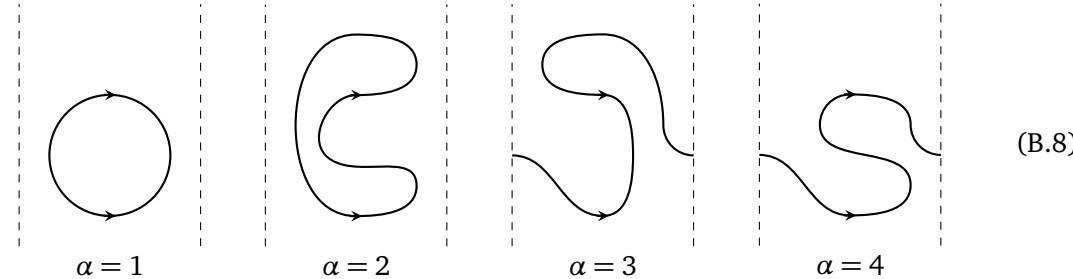

$$\alpha = 1 \qquad \alpha = 2 \qquad \alpha = 3 \qquad \alpha = 4 \tag{B.8}$$

We choose the boundary conditions to be empty (all sites are empty) at a time-like distance $M$ before (after) the insertion of the first (last) marked edge. For a chosen numerical precision of $N_{\text{digits}}$ decimal digits, in order to emulate the situation of an infinitely high cylinder, it suffices to take $M$ sufficiently large, so that the result $\widetilde{Z}_\alpha(K_c,n)$ does not depend on $M$ to within the chosen precision. We apply the extremely conservative choice $M = LN_{\text{digits}}$. As in [41] we take $N_{\text{digits}} = 2\,000$. In practice we can make the computations for $L \leq 5$ for several hundred different values of $t$ (we always take $t = 100, 101, \ldots$).

The computation of $\widetilde{Z}_\alpha(K,n)$ requires us to augment the state space of $\widetilde{T}$ with some extra information, which is not present in the unadorned link patterns on which $T$ acts. A crucial point is to keep this information at an absolute minimum, since it affects adversely the dimension of $\widetilde{T}$.

The key idea is to endow each arc of the link patterns with information about its possible interaction with the marked edges. An arc can be unmarked as before (meaning that it has passed through none of the two marked edges), or it can be simply or doubly marked (meaning that it has passed through one or both of the marked edges). A simply marked arc moreover has an orientation that specifies whether the edge, when followed from left to right (with respect to an observer who looks along the time direction), has passed through the marked edge along or opposite to the orientation of the marked edge. Obviously, there can be at most two simply marked arcs in a given state (in which case there are no doubly marked arcs), or one doubly marked arc (in which case there are no singly marked arcs).

The state moreover possesses an overall (i.e., not specific to each of its arcs) variable $\beta$ from which we will be able to infer to which $\widetilde{Z}_\alpha$ it will contribute at the end of the computation. This $\beta$ can take five different values, $\beta = 0, 1, 2, 3, 4$. The first one, $\beta = 0$, means that there is not yet any doubly marked arc, and therefore the contribution of the given state to the end result is yet undetermined. The other values, $\beta = 1, 2, 3, 4$, mean that the state gives a contribution to $\widetilde{Z}_\alpha$ with $\alpha = \beta$. More precisely, as soon as a doubly marked arc is formed, we set $\beta = 1$ or $2$, depending on whether the passage directions of the two marked edges are opposite or identical. Due to the choice of boundary conditions, the doubly marked arc must eventually close so as to form a loop. When this happens we keep the value $\beta = 1, 2$ is the marked loop is contractible, and we replace $\beta \mapsto 2 + \beta$ if it is non-contractible. In this way we correctly find $\alpha = \beta$ once the marked loop has been closed.

Keeping track of this information under the transitions produced by the transfer matrix $\widetilde{T}$ is quite non-trivial and a substantial number of cases needs to be accounted for. The salient features of this reasoning are the following:

1. The marking of arcs will add up upon concatenation of two distinct arcs by the cap operator. For example, the concatenation of a singly marked arc with an unmarked arc produces a singly marked arc, while the concatenation of two singly marked arcs produces a doubly marked arc.

2. However, the orientation of a singly marked arc may change when it is concatenated with an unmarked arc. Indeed, depending on how the two arcs are nested, in some cases an observer that follows the resulting concatenated arc from left to right will in fact follow the original singly marked arc from right to left. The orientation of the singly marked arc may therefore change upon concatenation, in some cases.

3. In a similar vein, when a doubly marked arc is formed by the concatenation of two singly marked arcs, the choice between $\beta = 1$ and $\beta = 2$ (see above) is determined by both the relative orientations of the singly marked arcs being concatenated and the way they are nested.

4. The cap operator should be prevented from forming a loop out of a singly marked arc, in order to account for the constraint that both marked edges must belong to the same marked loop.

The construction of $\widetilde{T}$ just outlined has undergone extensive tests on small lattices, comparing with an exhaustive set of diagrams drawn by hand.

## Results for the spectra

We now restrict to the critical coupling, $K = K_c$, given by (B.3). The following results are based on numerical computations for many values $n = \frac{1}{10}, \frac{3}{10}, \dots, \frac{19}{10}$ along both the dense and dilute branches.

We first focus on the decomposition of the probabilities

$$P_\alpha(n) = \frac{\widetilde{Z}_\alpha(K_c, n)}{Z(K_c, n)}, \tag{B.9}$$

for $\alpha = 1, 2, 3, 4$ onto the spectra $V_{\ell,k,m}$. We stress here that the probabilities $P_\alpha(n)$ are computed from $\widetilde{T}$, whereas the sets of eigenvalues $V_{\ell,k,m}$ are those of the simpler transfer matrix $T$. The amplitude of $P_\alpha(n)$ on each eigenvalue is found by solving a linear system using many different values of $t$ (see [27, 41] for technical remarks) and we are interested in the "spectrum" of each probability, meaning the set of eigenvalues for which the amplitudes are non-zero for generic values of $n$.

We draw our conclusions from the sizes $L = 3$ and $L = 4$, which are small enough that all amplitudes can be determined, and yet large enough to contain a substantial number of different sectors $(\ell, k, m)$, so that reliable conjectures about the general result can be made.

Our first observation is that the leading contributions to $\alpha = 1$ and $\alpha = 4$ are identical. This can be argued graphically, since once the doubly marked arc has been formed, it has almost the same probability to close on the front or the back of the cylinder. In the same vein, the leading contributions to $\alpha = 2$ and $\alpha = 3$ are identical. We therefore analyse the combinations

$$P_A^\pm = P_1(n) \pm P_4(n), \tag{B.10a}$$
$$P_B^\pm = P_2(n) \pm P_3(n). \tag{B.10b}$$

Our numerically determined amplitudes then unambiguously support the following conjectures:

- $P_A^+$ and $P_B^+$ both have contributions from $V_{\ell,k,m}$ with $\ell \geq 2$ even, $k$ always even, and any $m$.

- $P_A^-$ and $P_B^-$ both have contributions from $V_{\ell,k,m}$ with $\ell \geq 2$ even, $k$ having the same parity as $\ell/2$, and any $m$.

We can also form the combinations

$$P_{\text{all}} = P_A^+ + P_B^+, \tag{B.11}$$
$$P_{\text{Orient}} = P_A^- - P_B^- = P_1(n) - P_2(n) + P_3(n) - P_4(n). \tag{B.12}$$

The first one, $P_{\text{all}}$, is just the total probability that a loop passes through the two marked edges, regardless of any orientational information. This probability could of course have been obtained from a much simpler transfer matrix acting on link patterns without arc orientations, yet still having the arc markings. We find that $P_{\text{all}}$ has contributions only from $V_{2,0,m}$ (with any $m$), in full analogy with the study of two-point functions in [41]. The second one, $P_{\text{Orient}}$, exploits the full information of the decorated link patterns described above. It is the current-current correlation function argued in Section 6.3, and also coincides with the correlation function introduced by Cardy [2] in order to compute the area enclosed by the marked loop. We find that $P_{\text{Orient}}$ has contributions only from $V_{2,1,m}$ (with any $m$). In both these cases, $P_{\text{all}}$ and $P_{\text{Orient}}$, there are *no* contributions from any $V_{\ell,k,m}$ with $\ell > 2$.

Summarizing, $P_{\text{all}}$ couples to the two-leg operator, while $P_{\text{Orient}}$ couples to the current—and to nothing else, in both cases (in particular, there are no contributions from sectors with $\ell > 2$).

**Results for the amplitudes**

For $-2 < n \le 2$, using (1), the Coulomb gas coupling constant $\beta^2$ takes the following values:

$$\beta^2 = \begin{cases} \frac{1}{\pi}\arccos\left(-\frac{n}{2}\right) \in (0,1], & \text{Dense phase,} \\ 2 - \frac{1}{\pi}\arccos\left(-\frac{n}{2}\right) \in [1,2), & \text{Dilute phase.} \end{cases} \tag{B.13}$$

where the central charge is given (1). While Cardy's prediction for the current-current amplitude is given by eq. (27) in [9]:

$$\kappa_C(\beta^2) = \frac{2(\beta^2 - 1)}{\pi \beta^2}\cot(\pi\beta^2). \tag{B.14}$$

We now wish to relate the probability $P_{\text{Orient}}$ found in the lattice computations to $\kappa_C(\beta^2)$. Due to the normalization (B.9), the amplitude $A(L, \beta^2)$ of its leading contribution—namely the largest eigenvalue in $V_{2,1,m}$—is well defined. Notice that in our numerics we do not distinguish states which only differ by the sign of the lattice momentum, since the corresponding transfer-matrix eigenvalues are exactly degenerate for finite $L$. Recall that the lattice momentum carries over as $\Delta - \bar\Delta$ in the CFT. When extracting the lattice amplitude of a spinful field—such as the current—we therefore let $A(L, \beta^2)$ denote only one half of the combined amplitude of the two degenerate eigenvalues.

We cannot compare $A(L, \beta^2)$ and $\kappa_C(\beta^2)$ directly, because the former is computed in the cylinder geometry, while the CFT result pertains to the infinite plane. We study instead the conformal amplitude $A(L, \beta^2)$:

$$A_{\text{CFT}}(L, \beta^2) \equiv \left(\frac{2\pi}{\frac{\sqrt{3}}{2}L}\right)^{-2} A(L, \beta^2), \tag{B.15}$$

where the geometrical factor $\sqrt{3}/2$ corrects for the fact that on a hexagonal lattice our discretisation of the time and space coordinates differ by a factor that equals the height of an equilateral triangle of side length one. The power 2 is twice the scaling dimension of the current operator.

On a practical level, the fact that we now need only the leading amplitude means that we need only a few values of the separation $t$, and we can moreover work at a much smaller numerical precision. Concretely we determine the first five amplitudes, using 100 digits of numerical precision, which is amply sufficient to determine $A(L, \beta^2)$ to at least 20 correct digits. As a consequence, the computations can now be carried out for sizes $L \le 6$.

In figure 1 we show $A_{\text{CFT}}(L, \beta^2)$, plotted against $n$ on the dense and dilute branches, for sizes $L = 3$ (red), $L = 4$ (blue), $L = 5$ (green) and $L = 6$ (orange). We also show various extrapolations, using a second-order polynomial in $1/L$ for sizes $L = 3, 4, 5$ (grey) and $L = 4, 5, 6$ (magenta), or a third-order polynomial for all sizes $L = 3, 4, 5, 6$ (purple). Finally, the quantity $\frac{1}{16}\kappa_C(\beta^2)$ is shown as a yellow dashed curve. The latter analytical result is nicely framed by the latter two extrapolations in a band of around 1% for all values of $n$.

Our numerical result thus confirms that

$$P_{\text{Orient}} = \left(\frac{4\pi}{\sqrt{3}L}\right)^2 \frac{\kappa_C}{16}\left[\left(\frac{1}{2\sinh\frac{2\pi}{\sqrt{3}L}w}\right)^2 + \text{h.c.}\right], \tag{B.16}$$

where $w = \sigma + i\tau$ are complex coordinates on the cylinder. From (B.16) we deduce that, at small distance

$$P_{\text{Orient}} = \frac{\kappa_C}{16}\left(\frac{1}{w^2} + \text{h.c.}\right). \tag{B.17}$$

On the other hand, with the definition we have used in the text, $P_{\text{Orient}} = \langle j_\mu j_\mu \rangle$ in (108). This confirms that $\kappa = \frac{\kappa_C}{16}$, where $\kappa$ is the constant used in the main text and given by (109).

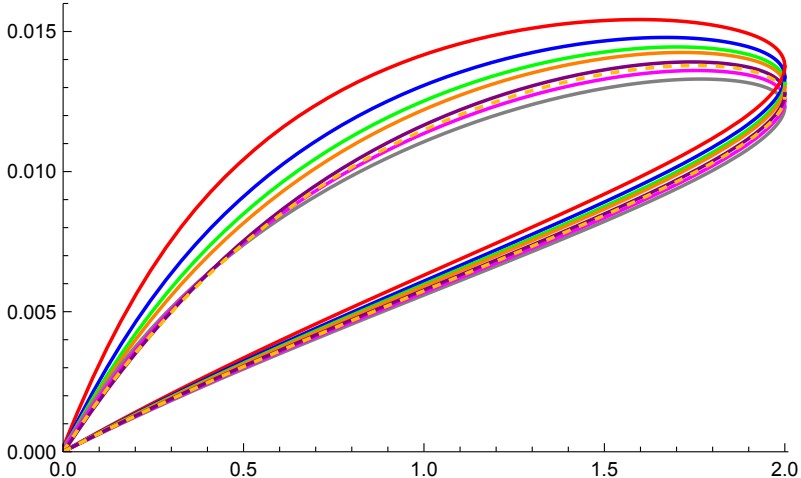

Figure 1: Comparison between $A_{\text{CFT}}(L, \beta^2)$ and the analytical result $\frac{1}{16}\kappa_{\text{C}}(g)$.

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
