# Peer review of "On currents in the $O(n)$ loop model"

_SciPost Physics, doi:SciPost Phys. 16, 111 (2024)_

## Round 1 · Referee Report · Anonymous (Referee 1) · 2024-1-22

Report
This paper analyzes CFT data about Noether currents in critical $O(n)$ loop model using methods of numerical Virasoro conformal bootstrap. An important contribution of the paper is a bootstrap study of problems which is usually tackled using Coulomb gas calculations. In section $2$, the authors study current-current OPEs and show that there is no underlying Kac-Moody algebra that would have followed in a unitary $2D$ critical system with continuous symmetry, described by Wess-Zumino-Witten CFT in the continuum. In section $3$ the authors continue their study of $JJ$ OPE using analytical approach as well as numerical bootstrap. They show outstanding agreement between the results obtained from the two methods. Then in sections $4$ the authors study current-current OPE in the $n\to\pm 2$ free-field limits of the loop model. In section $5$ general $n$ loop models, like self-avoiding random walk, are investigated. This paper is an interesting application of numerical conformal bootstrap adapted to non-unitary CFT with important physical implications for $2D$ critical loop models. This work is also important in comparing numerical bootstrap results with results obtained from other non-bootstrap analytical methods like the Coulomb gas technique.
I recommend the publication of this paper after the authors address the following points and incorporate the changes requested.
$\textbf{Requests for clarification}$
-
In equation (3.8) the authors have introduced the structure constants $C_{JJV}^{\rm ref}, \, C_{VJJ}^{\rm ref}$ which they have called the $\textit{reference structure constants}$. Is there any particular reason why these are called with the prefix $\textit{reference}\, ?$ What are their relations to the structure constants $C_{JJV}$ introduced in prior discussions, like, for example, in eq. (2.7)?
-
Eq (3.16) gives conjectural upper bounds on the degrees of the polynomial $q^{\Lambda}_{(r,s)}(n)$ for $r\le 5$ based on the analysis of a companion paper cited as ref [25].
A) What is so special about $r=5$? Is there a reason why there is no conjecture for $r>5$?
B) The authors provide explicit expressions for various $q^\Lambda_{(r,s)}(n)$ in equations (3.18)-(3.21). The polynomials in equations (3.18) and (3.19) saturate the respective upper bounds on the degree of the polynomials. Are there other polynomials of the same class which does not saturate the upper bounds given in eq(3.16), like some of the polynomials in (3.20)-(3.21) which do not saturate the respective upper bounds? Since, in the preceding paragraph, the authors claim that these polynomials can be uniquely determined by the crossing-symmetry equation, is there a way to analyze the crossing-symmetry equation to predict whether the polynomial will saturate the upper bound of the degree?
-
Are there particular reason why the authors consider $\langle\bar{J}\bar{J}\bar{J}\bar{J}\rangle$ instead of $\langle JJJJ\rangle$ for numerical bootstrap?
$\textbf{Requests for changes}$
$\S$ The review paragraph on the spectrum of $O(n)$ CFT is a natural entry point into the paper's analysis. Starting section 2 with this review paragraph, instead of relegating it to the end of the introduction, sets a better flow.
$\S$ The authors claim that the various equalities in equation (3.17) can be established using the degenerate shift equation (2.7). Since this does not seem obvious, and this provides the logical precedence to the subsequent numerical analysis, it is desirable to have this worked out explicitly. At the least, the authors are requested to provide a clear flow of logic as to how one can show these equalities.
$\S$ Since this paper uses important techniques, both analytical and numerical, some short reviews of these will improve the readability of the paper. In particular, a gist of the results of ref [25], which is cited time and again throughout the paper, is needed to understand some important points that have been raised in point $3$ above. Further, a review of the basic setup for the numerical bootstrap, like some formal expressions for the various truncated crossing-symmetry equations, is needed to understand how the numerical scheme is used.
$\S$ Please use some different colors in the diagrams of (A.28) in Appendix A which can act as a better guide for the arguments presented in the preceding paragraphs.
$\S$ Please update the ref [25].

---

## Round 1 · Referee Report · Anonymous (Referee 2) · 2024-2-2

Report
In the submitted manuscript the authors have considered 2d CFTs with a $O(n)$ symmetry which describe the critical point of $O(n)$ loop models. They work with the current operators $J, \bar{J}$ and show the spectrum of operators in their OPEs. It is pointed out from the expressions of the OPEs that the currents are non-holomorphic and do not have a Kac-Moody algebra. In sec 3, they show how the 4-point function $\langle JJJJ\rangle$ is determined by the spectrum up to some structure constants i.e. OPE coefficients. These structure constants are determined from a bootstrap analysis and the results are compared with an analytic formula, which is obtained in a companion paper. This analytic formula is also used to determine the level parameter $k$ i.e. the residue of the leading singularity in the $J(x)J(0)$ OPE.
In sec 4, the special cases of $n=\pm 2$ are discussed separately. In the final section the authors have argued that the marginal operator $J\bar{J}$ decouple from the theory and hence cannot be a deformation of the critcal loop model as argued in ref. [1] of the paper. In this context they discuss the two point function of orientation of loops in a complex $O(n/2)$ model, which depends on the parameter $k$. The expression for $k$ obtained in the paper matches that obtained in refs. [2,9] using Coulomb gas technique.
I find the analysis and results of this paper quite robust and interesting. I would recommend it for publication, but I request the authors for clarifications on the following two points:
-
For the bootstrap analysis in sections 3.1 and 3.2, it is argued that an infinite tower of Virasoro primaries can be packed together into a single interchiral conformal block. It is then convenient to truncate the spectrum of primaries in the crossing equation. However it was not clear to me how the truncation was done. Was the complete interchiral block computed so that the truncation was only in the first primary in the tower? Or was the interchiral block itself truncated at some order of expansion?
-
It is clear that the bootstrap analysis was performed for an arbitrary $n$ and the result matched with the analytic expression (3.8). However is it not possible to have some special value of $n$ where there are other crossing symmetric solutions, that cannot be captured in the analysis of sec. 3.2? Is there any reason to rule out such solution? If not, should eq (3.8) be modified for such $n$'s?
I also noticed that one of the authors of ref. [9] was not credited in the paragraph above eq. (5.19).

---

## Round 2 · Referee Report · Anonymous (Referee 2) · 2024-3-30

Report

I am satisfied with the changes in the revised version, and the responses to my previous questions/comments. I recommend the paper for publication.

---

## Round 2 · Referee Report · Anonymous (Referee 1) · 2024-3-30

Report

The authors have successfully addressed the clarification requests. I am also satisfied by the modifications incorporated into the new draft. Thereby I happily recommend publication of the work in its latest form.

---

## Round 2 · Author Response

We have made adjustments according to requests and comments of the two referees

---

## Round 2 · List of Changes

Changes according to the report 1:

  1. We have added a section for the review on the spectrum of the $O(n)$ CFT as Section 2.

  2. We have sketched the derivation for the identities of $q_{r,s}$ in (4.17) [3.17 in the old version]. See the details between the equation (4.17) and (4.19). 

The main idea here is that the ratios of structure constants from four-point functions of $J$ and $\bar J$ in (4.19) are completely determined by the degenerate-shift equation, which also coincides with the ratios of the corresponding reference structure constants.

  1. We have summarized the main ideas of the numerical bootstrap techniques required for our results as bullet points on page 18.

  2. We have added different colors to different $O(n)$ labels on diagrams in (A.28).

  3. We have updated reference [25]. 
 Clarifications according to the report 1:

  4. What we call reference structure constants are universal factors of structure constants that are independent of model’s global symmetry, that is to say reference structure constants serve as references for structure constants of primary fields with the same dimensions, but transform in different $O(n)$ representations. 

We have added the above clarification above the equation (4.14a).

From our results (4.8), we expect that the three-point functions $C_{JJV}$ is a product between polynomials in $n$ and the reference structure constants, as written in (4.24). This expectation seems to make sense because the model’s symmetry is, a priori, a product between $O(n)$ and conformal symmetry.

  1. The equation (4.16) [3.16 in the old version] is a conjecture for the bounds of the polynomials’ degree for any value of $r,s$: this has been now stressed above (4.16), however we have only explicitly checked the inequalities in (4.16) for $r<=5$.

In general, there are certainly polynomials that do not saturate the bounds (4.16). Unfortunately, our results on the polynomials and their degrees were obtained based only on the numerical observations, and we do not yet know how to determine their degrees by analyzing the crossing-symmetry equation.

Above the title of Section 4.2, we have added a paragraph to stress that our results on the polynomials were obtained based on purely numerical observations.

  1. We have considered $<\bar J\bar J\bar J\bar J>$ instead of $<JJJJ>$ due to purely technical reasons.

Our numerical bootstrap program, initially proposed in the paper (*), was designed for four-point functions of primary fields with positive Kac indices. Recall that $\bar J$ carries the Kac indices $(1,1)$ whereas we have $(1,-1)$ for $J$.

In practice, interchiral blocks of those two four-point functions take slightly different forms due to the singularities in the degenerate-shift equations and logarithmic blocks, and the details on how to regularize those singularities is given in Section (3.1) of the paper (*). And we chose to write the program that fits with the case $<\bar J\bar J\bar J\bar J>$

However, since structure constants in $<\bar J\bar J\bar J\bar J>$ and $<JJJJ>$ are related by the degenerate-shift equation, it is enough to consider only 1 of them, and we chose to consider $<\bar J\bar J\bar J\bar J>$

*=https://arxiv.org/pdf/2111.01106.pdf

Clarifications according to report 2:

  1. In each interchiral block, we apply the truncation to any fields, both primaries and descendants, such that the remaining fields obey our desired bound. We have stressed this point in the first bullet point on page 18.

That is to say, we truncate each interchiral block to be a finite sum of truncated Virasoro blocks wherein we only include the contribution from the descendant fields up to some certain levels.

  1. In practice, it is more convenient to discuss the analyticity of (4.8) in terms of the parameter $\beta^2$ because four-point functions of the $O(n)$ CFT depends explicitly on $\beta^2$. Recall that $\beta^2$ is related to $n$ through equation (2.1).

The analytic structure constants (4.8) [3.8 in the old version] are only valid for non-rational value of $\beta^2$ because (4.8) could have zeros and poles at rational value of $\beta^2$. The latter case also includes some integers $n$, for instance $\beta^2 = 1$ corresponds to $n=2$. 

We expect that crossing symmetric solutions for rational $\beta^2$ can be obtained as rational limits of (4.8) wherein we expect that those poles in rational $\beta^2$ always cancel each others. 

While we do not know yet the complete mechanism of pole cancellation, we believe that this should be true because the four-point functions in the $O(n)$ loop model on the finite-size lattice exist for generic $n$, including $\beta^2$.

We have added the above clarification in the first paragraph of page 14.

---

## Editorial Decision

published